# Content-specific activity in frontoparietal and default-mode networks during prior-guided visual perception

Carlos González-García[1,2], Matthew W Flounders[1,3†], Raymond Chang[1†], Alexis T Baria[1], Biyu J He[1,3,4,5,6*]

[1]National Institute of Neurological Disorders and Stroke, National Institutes of Health, Bethesda, United States; [2]Department of Experimental Psychology, Ghent University, Ghent, Belgium; [3]Neuroscience Institute, New York University Langone Medical Center, New York, United States; [4]Departments of Neurology, New York University Langone Medical Center, New York, United States; [5]Departments of Neuroscience and Physiology, New York University Langone Medical Center, New York, United States; [6]Departments of Radiology, New York University Langone Medical Center, New York, United States

**Abstract** How prior knowledge shapes perceptual processing across the human brain, particularly in the frontoparietal (FPN) and default-mode (DMN) networks, remains unknown. Using ultra-high-field (7T) functional magnetic resonance imaging (fMRI), we elucidated the effects that the acquisition of prior knowledge has on perceptual processing across the brain. We observed that prior knowledge significantly impacted neural representations in the FPN and DMN, rendering responses to individual visual images more distinct from each other, and more similar to the image-specific prior. In addition, neural representations were structured in a hierarchy that remained stable across perceptual conditions, with early visual areas and DMN anchored at the two extremes. Two large-scale cortical gradients occur along this hierarchy: first, dimensionality of the neural representational space increased along the hierarchy; second, prior's impact on neural representations was greater in higher-order areas. These results reveal extensive and graded influences of prior knowledge on perceptual processing across the brain.
DOI: https://doi.org/10.7554/eLife.36068.001

**\*For correspondence:**
biyu.jade.he@gmail.com

[†]These authors contributed equally to this work

**Competing interests:** The authors declare that no competing interests exist.

## Introduction

Prior experiences can have an enormous impact on perception. For example, a harmless rope on the trail may be perceived as danger and trigger a jump in fright if one has had a recent encounter with a snake. Although it is well established that perception arises as a consequence of a continuous interaction between incoming sensory input and internal priors (*Bastos et al., 2012*; *Mumford, 1992*; *Yuille and Kersten, 2006*; *Albright, 2012*; *Summerfield and de Lange, 2014*; *Trapp and Bar, 2015*), the neural mechanisms underlying this process remain unclear. In particular, while extensive research has focused on how prior experience and knowledge shape neural processing in early sensory areas (e.g., [*Kok et al., 2012*; *Schlack and Albright, 2007*; *Alink et al., 2010*; *Hsieh et al., 2010*]), whether higher-order frontoparietal regions contain content-specific neural representations that are involved in prior-guided perceptual processing remains unknown.

Here, we test whether content-specific neural activity exists during prior-guided visual processing in the default-mode network (DMN) (*Raichle et al., 2001*) and frontoparietal network (FPN) (*Dosenbach et al., 2008*). Classic theories suggest that the DMN is involved in internally oriented processes (*Buckner et al., 2008*), such as self-related processing (*Kelley et al., 2002*), spontaneous/

task irrelevant thought (*Mason et al., 2007*; *Christoff et al., 2009*), episodic and semantic memory (*Shapira-Lichter et al., 2013*; *Binder et al., 2009*; *Sestieri et al., 2011*). This idea is challenged by recent studies showing DMN activation in externally oriented tasks, including executive control and working memory tasks (*Crittenden et al., 2015*; *Konishi et al., 2015*; *Vatansever et al., 2015*). These results have inspired a recent framework suggesting that the DMN, being positioned farthest away from uni-modal sensory and motor areas in both functional connectivity and anatomical space, allows online information processing to be guided by stored representations (*Margulies et al., 2016*). However, to date evidence for content-specific activity in the DMN during externally-oriented tasks is still lacking, which would be crucial for establishing this network's role in specific computations linking online processing with stored representations. We reasoned that prior-guided sensory processing, which requires using previously acquired priors (broadly defined) to guide online perceptual processing, is an ideal context for investigating this possibility. In particular, the role of DMN in prior-guided visual perception remains controversial (*Dolan et al., 1997*; *Gorlin et al., 2012*).

Whether the FPN is involved in content-specific perceptual processing is also unclear. One framework suggests that this network only encodes task/response-relevant information (*Erez and Duncan, 2015*; *Bracci et al., 2017*), such as learnt arbitrary categories of visual objects that are relevant to performance (*Freedman et al., 2001*) or the distinction between a target and a non-target (*Erez and Duncan, 2015*). On the other hand, neural activity reflecting perceptual content has been observed in both the lateral prefrontal cortex (LPFC)(*Wang et al., 2013*; *Panagiotaropoulos et al., 2012*; *Mendoza-Halliday and Martinez-Trujillo, 2017*) and posterior-parietal cortex (PPC) (*Jeong and Xu, 2016*; *Konen and Kastner, 2008*; *Freud et al., 2016*) components of this network, even when task demand is held constant or was irrelevant to the perceptual content (e.g., [*Panagiotaropoulos et al., 2012*; *Mendoza-Halliday and Martinez-Trujillo, 2017*; *Konen and Kastner, 2008*]). Currently, whether the LPFC contributes to visual perceptual processing is a heavily debated topic (*Dehaene et al., 2017*; *Koch et al., 2016*). Inspired by the observation that in anatomical and functional connectivity space, the FPN is situated in between the DMN and sensory areas (*Margulies et al., 2016*), we reasoned that the FPN may exhibit an intermediate pattern of effects compared to the DMN and sensory areas during prior-guided perceptual processing.

In the laboratory, Mooney images provide a well-controlled paradigm for studying how prior experience shapes perceptual processing. Mooney images are black-and-white degraded images that are difficult to recognize at first. Yet, once the subject is exposed to the original, non-degraded grayscale image (a process called 'disambiguation'), their recognition of the corresponding Mooney image becomes effortless, and this effect can last for days to months, even a lifetime (*Albright, 2012*; *Ludmer et al., 2011*). This phenomenon demonstrates that experience can shape perception in a remarkably fast and robust manner.

The Mooney images paradigm thus allows comparing neural processing of a repeatedly presented, physically identical image that leads to distinct perceptual outcomes depending on whether a prior due to earlier experience is present in the brain. Recent studies have shown that after disambiguation, neural activity patterns elicited by Mooney images in early and category-selective visual regions become more distinct between different images and more similar to those elicited by their matching grayscale counterparts (*Hsieh et al., 2010*; *Gorlin et al., 2012*; *van Loon et al., 2016*). However, how prior experience impacts neural processing in higher-order frontoparietal cortices remains largely unknown. In this study, we investigated neural processing of Mooney images before and after disambiguation across the brain, from retinotopic and category-selective visual areas to FPN and DMN, using ultra-high-field (7T) fMRI and multivariate pattern analyses. We hypothesized that the acquisition of perceptual priors would alter neural representations and enhance content-specific information in DMN and FPN regions.

## Results

### Task paradigm and behavioral results

Nineteen subjects were shown 33 Mooney images, 17 of which contained animals, and 16 contained manmade objects. Each Mooney image was presented six times before its corresponding gray-scale image was shown to the subject, and six times after. Following each Mooney image presentation, subjects responded to the question 'Can you recognize and name the object in the image?' using a

button press ('subjective recognition'). Each fMRI run included three distinct gray-scale images, their corresponding post-disambiguation Mooney images, and three new Mooney images shown pre-disambiguation (their corresponding gray-scale images would be shown in the next run), with randomized order within each stage such that a gray-scale image was rarely followed immediately by its corresponding Mooney image (*Figure 1A*; for details, see Materials and methods, *Task paradigm*). To ensure that subjects' self-reported recognition matched the true content of the Mooney images, at the end of each run, Mooney images presented during that run were shown again and participants were asked to verbally report what they saw in the image and were allowed to answer 'unknown'. This resulted in a verbal test for each Mooney image once before disambiguation and

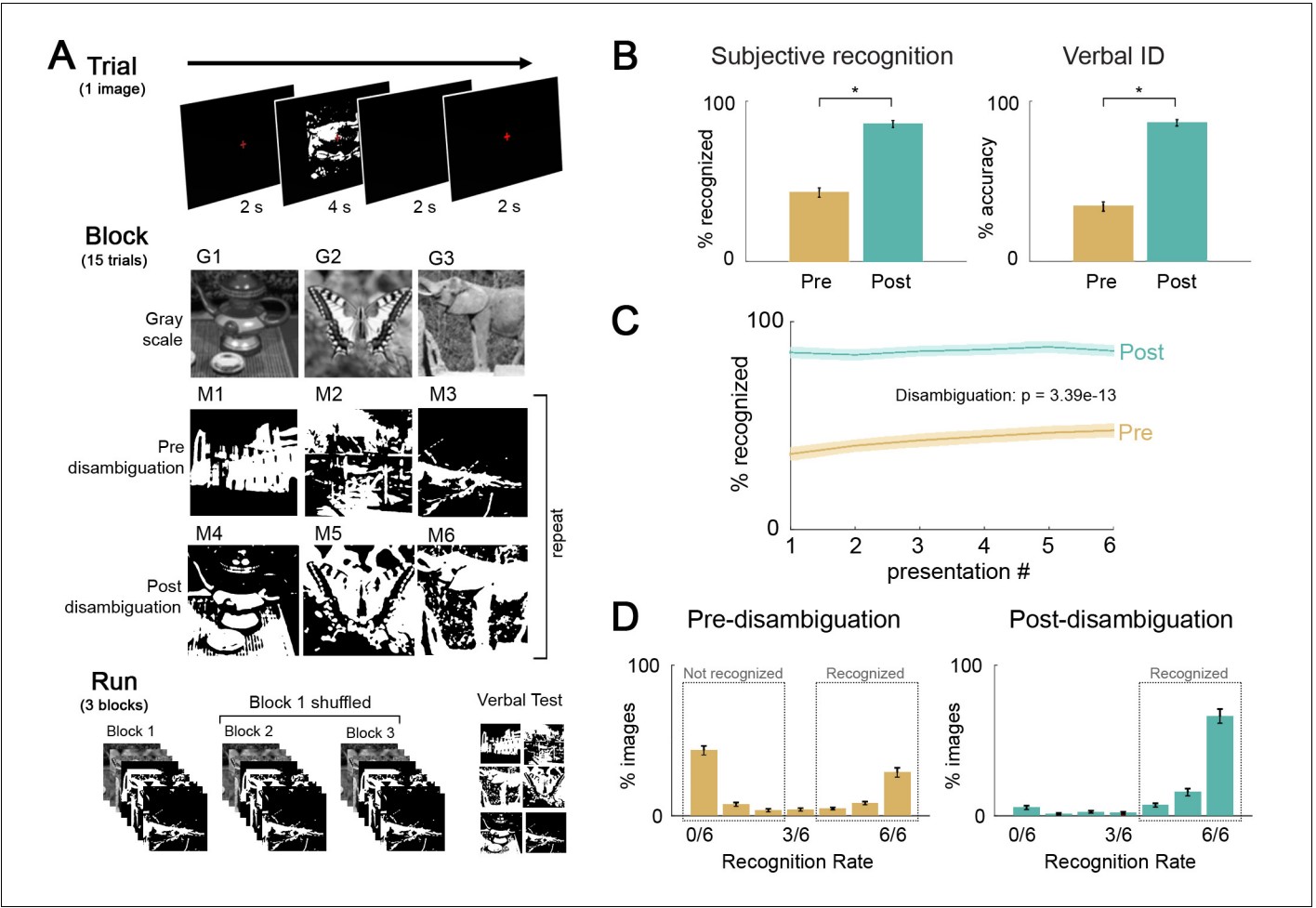

**Figure 1.** Paradigm and behavioral results. (**A**) Task design, and flow of events at trial, block, and fMRI run level. Subjects viewed gray-scale and Mooney images and were instructed to respond to the question 'Can you recognize and name the object in the image?'. Each block included three gray-scale images, three money images corresponding to these gray-scale images (i.e., post-disambiguation Mooney images), and three Mooney images unrelated to the gray-scale images (i.e. pre-disambiguation Mooney images, as their corresponding gray-scale images would be presented in the following block). 33 unique images were used and each was presented six times before and six times after disambiguation (see Materials and methods for details). (**B**) Left: Percentage of 'recognized' answers across all Mooney image presentations before and after disambiguation. These two percentages significantly differed from each other (p=3.4e-13). Right: Percentage of correctly identified Mooney images before and after disambiguation. These two percentages significantly differed from each other (p=1.7e-15). (**C**) Recognition rate for Mooney images sorted by presentation number, for the pre- and post-disambiguation period, respectively. A repeated-measures ANOVA revealed significant effects of the condition (p=3.4e-13), the presentation number (p=0.002), and the interaction of the two factors (p=0.001). (**D**) Distribution of recognition rate across 33 Mooney images pre- (left) and post- (right) disambiguation. Dashed boxes depict the cut-offs used to classify an image as recognized or not-recognized. All error bars denote s.e.m. across subjects.

DOI: https://doi.org/10.7554/eLife.36068.002

once after disambiguation ('verbal identification'). Verbal responses were scored as correct or incorrect using a pre-determined list of acceptable responses for each image.

Disambiguation by viewing the gray-scale images had a substantial effect on participants' subjective recognition responses, with significantly higher rate of recognition for Mooney images presented post-disambiguation (86 ± 1%; mean ± s.d. across subjects) compared to the same images presented before disambiguation (43 ± 12%; $t_{1,18}$ = 18.6, p=3.4e-13, Cohen's d = 4.2; *Figure 1B*, left). A similar pattern of results was observed using the verbal identification responses. Mooney images were correctly identified significantly more often after disambiguation (86 ± 0.8%) than before (34 ± 12%; $t_{1,18}$ = 25.3, p=1.7e-15, Cohen's d = 5.8; *Figure 1B*, right).

A two-way ANOVA on recognition rate with presentation number (*Bastos et al., 2012*; *Mumford, 1992*; *Yuille and Kersten, 2006*; *Albright, 2012*; *Summerfield and de Lange, 2014*; *Trapp and Bar, 2015*) and disambiguation stage (pre- vs. post-) as factors revealed that disambiguation had a dramatic effect on recognition rate ($F_{1,18}$ = 345.6, p=3.4e-13, $\eta^2_p$ =.95, while repetition also improved recognition ($F_{5,90}$ = 8.1, p=0.002, $\eta^2_p$ = 0.3) (*Figure 1C*). The interaction between these two factors was significant ($F_{5,90}$ = 7.5, p=0.001, $\eta^2_p$ = 0.29). A post-hoc analysis (Bonferroni-corrected) suggested that the repetition effect was driven by pre-disambiguation images, with recognition rate increasing gradually across presentations before disambiguation (p<0.05), but not after disambiguation (p=1). For instance, for pre-disambiguation images, the recognition rate was significantly higher in the sixth presentation (47 ± 14%, mean ± s.d. across subjects) compared to the first presentation (36 ± 13%; paired t-test across subjects, $t_{1,18}$ = 4.2, p=0.007, Cohen's d = 0.97), suggesting that participants did not stop attending after being unable to recognize Mooney images at first, and actively tried to recognize these in subsequent presentations.

Based on the bimodal distribution of subjective recognition rates across images (*Figure 1D*), we established cut-offs to select 'pre-(disambiguation) not-recognized' images as those recognized two or fewer times before disambiguation, 'pre-(disambiguation) recognized' images as those recognized at least 4 out of 6 times before disambiguation, and 'post-(disambiguation) recognized' images as those recognized four or more times after disambiguation. Accuracy of verbal identification responses in these three groups were 8.7 ± 5.8%, 70.6 ± 16.0%, 93.6 ± 4.5% (mean ± s.d. across subjects), respectively. Due to the low number of images, the category 'post-(disambiguation) not-recognized' was not included in further analyses. Results described below using subjective recognition reports based on these cut-offs are very similar to those obtained using correct vs. incorrect verbal identification responses.

## Disambiguation leads to widespread changes in neural activity

We first investigated which brain regions encode the *status* of a Mooney image (pre- vs. post- disambiguation) in their overall activation magnitudes or voxel-wise activity patterns, using general linear model (GLM) and multivariate pattern analysis (MVPA), respectively. For this analysis we selected, for each subject, the set of Mooney images that were both unrecognized in the pre-disambiguation stage and recognized in the post-disambiguation stage. First, consistent with earlier studies (*Dolan et al., 1997*; *Gorlin et al., 2012*), we found that disambiguated Mooney images elicited significantly higher activity in regions of the DMN, including the posterior cingulate cortex (PCC), lateral parietal cortices (Par Lat), and the medial prefrontal cortex (MPFC) than the same images presented pre-disambiguation (*Figure 2A*, p<0.05, FWE-corrected; results using verbal identification responses are similar and presented in *Figure 2—figure supplement 1*). A more detailed analysis revealed that this effect resulted from a reduced deactivation in the post-disambiguation period (*Figure 2B*, p<0.05, FWE-corrected). Using a searchlight decoding analysis across the whole brain, we found that information about the status of a Mooney image was contained in the voxel-wise activity pattern in an extensive set of brain regions (*Figure 2C*, p<0.01, corrected by cluster-based permutation test; results using verbal identification responses are similar and presented in *Figure 2—figure supplement 2*), comprising mainly the FPN (*Dosenbach et al., 2008*), but also including the anterior cingulate cortex (ACC), bilateral anterior insulae, fusiform gyri (FG), and components of the DMN.

There are several alternative interpretations of the above findings. The disambiguation-related activation contrast found in DMN regions could reflect perceptual priors present following disambiguation, as previously suggested (*Dolan et al., 1997*). Alternatively, these regions may lack encoding of image-specific information, and the changes in their activity magnitudes may reflect nonspecific effects such as increased arousal (*Gorlin et al., 2012*) or decreased task difficulty (*Singh and*

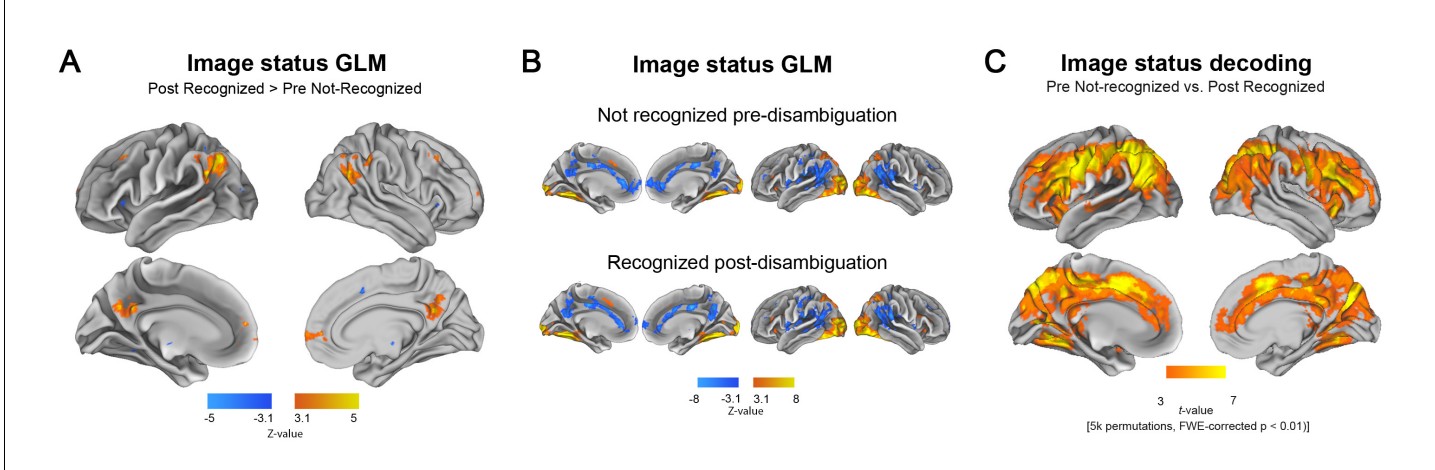

**Figure 2.** Disambiguation-induced changes in neural activity magnitude and pattern. (**A**) GLM contrast related to the disambiguation effect. For each subject, the set of Mooney images that were not recognized in the pre-disambiguation period and recognized in the post-disambiguation period are used. Thus, the GLM contrast is between an identical set of images that elicit distinct perceptual outcomes. Warm colors show regions with significantly higher activity magnitudes after than before disambiguation (p<0.05, FWE-corrected). (**B**) Activation and deactivation maps for each condition separately (p<0.05, FWE-corrected). Top row: Activation/deactivation map corresponding to pre-disambiguation, not recognized Mooney images, as compared to baseline. Bottom row: post-disambiguation, recognized Mooney images. (**C**) Searchlight decoding of Mooney image status (pre-disambiguation not-recognized vs. post-disambiguation recognized). For each subject, the same set of Mooney images are included in both conditions. Results are shown at p<0.01, corrected level (cluster-based permutation test).
DOI: https://doi.org/10.7554/eLife.36068.003

The following figure supplements are available for figure 2:

**Figure supplement 1.** GLM results of the *post-disambiguation identified >pre disambiguation not-identified* contrast, based on the verbal identification responses.
DOI: https://doi.org/10.7554/eLife.36068.004

**Figure supplement 2.** Searchlight decoding of *pre-disambiguation not-identified* vs *post-disambiguation identified* Mooney images, selected based on verbal identification responses.
DOI: https://doi.org/10.7554/eLife.36068.005

**Figure supplement 3.** Regions of interest (ROIs) used in RSA.
DOI: https://doi.org/10.7554/eLife.36068.006

**Figure supplement 4.** The definition of FPN ROIs from image status decoding results (for details, see Materials and methods, *ROI definition*).
DOI: https://doi.org/10.7554/eLife.36068.007

*Fawcett, 2008*) in the post-disambiguation period. Similarly, regions showing significant decoding of the disambiguation effect may respond to these nonspecific effects or, alternatively, contain content-specific activity related to the perceptual processing of the images. To adjudicate between these alternatives and determine whether DMN and FPN contain content-specific neural representations, we extracted four regions of interests (ROIs) from the GLM result shown in *Figure 2A*, including MPFC, PCC, and the left and right lateral parietal cortices (Par Lat), and four ROIs that represent key components of the FPN (bilateral frontal and parietal cortices) from the decoding result shown in *Figure 2C*. For each ROI, we used representational similarity analysis (RSA) to probe the neural representation format of Mooney images before and after disambiguation. We further defined retinotopic visual ROIs (from V1 to V4), lateral occipital complex (LOC) and fusiform gyrus (FG) for each subject using separate localizers to serve as a comparison to frontoparietal areas (for ROI definitions and replication with independent FPN and DMN ROIs, see Materials and methods, *ROI definition*, *Figure 2—figure supplement 3*, and *Figure 2—figure supplement 4*).

## Mooney images are encoded more distinctly after disambiguation throughout the cortical hierarchy

RSA sheds light onto the neural representational format by quantifying similarities or differences between neural representations of different images, or of the same image in different conditions (*Kriegeskorte et al., 2008*). For each subject and ROI, we obtained a representational dissimilarity

matrix (RDM) that contained the correlational distance (1 – Pearson's r) between every pair of images (*Figure 3A*; the RDM is a symmetrical matrix). Thirty-three Mooney images shown in the pre- and post-disambiguation period, and their grayscale counterparts, were arranged along the x and y axis of the matrix (*Figure 3A*). Correlational distance was then computed between every pair of images that were either in the same condition (green triangles in *Figure 3A*, top-right panel), or in different conditions (yellow squares in *Figure 3A*, bottom-right panel).

Strikingly, disambiguation increased representational dissimilarity between individual images in all ROIs investigated, including FPN and DMN. This can be seen from the gradient of color difference between the Pre-Pre square and the Post-Post square in the RDM (*Figures 3B* and *4A*, which include mid-line and right hemisphere ROIs; see *Figure 3—figure supplement 1* and *Figure 4—figure supplement 1* for similar results from the left hemisphere ROIs, and *Figure 3—figure supplement 2* for similar results from V4). The similarity structure of activation patterns for different images can be visualized by applying multi-dimensional scaling (MDS) to the RDM (see *Figure 3C and 4B* for 2-D MDS plots; SI Result for interactive 3-D MDS plots). In the MDS plot, each dot represents one image shown in a particular condition, and distances between dots preserve the correlational distance in the RDM as much as possible. As can be seen from the MDS plots, post-disambiguation Mooney images are represented more distinctly from each other (shown as greater distances among yellow and green dots) than their pre-disambiguation counterparts (blue dots), which are more clustered. This effect was most pronounced in FPN and DMN regions. Similarly, in all ROIs, grayscale images are represented relatively distinctly from each other (as shown by the large distances among red-pink dots).

To statistically assess this effect, we averaged the correlational distance across image pairs within each condition and compared the mean dissimilarity between conditions (*Figure 3A*, top-right panel). Representational dissimilarity between individual Mooney images significantly increased after disambiguation in all ROIs investigated across both hemispheres, except the left V2 and left V4 (*Figure 3D and 4C*, cyan brackets, all p<0.05, FDR-corrected, assessed by paired Wilcoxon signed-rank test across subjects; and see panel C of *Figure 3—figure supplement 1*, *Figure 4—figure supplement 1*, *Figure 3—figure supplement 2*), suggesting that Mooney images are represented more distinctly after disambiguation throughout the cortical hierarchy. In addition, representational dissimilarity between different images within each condition (shown as the height of bars in *Figure 3D and 4C*) gradually increased from early visual areas to category-selective visual regions, to FPN and DMN. A control analysis ruled out the possibility that this result was affected by different sizes of the ROIs, as equating ROI sizes yielded similar results (*Figure 4—figure supplement 2* and *Figure 3—figure supplement 2*).

Next, we assessed the amount of image category (natural vs. manmade) information in each ROI under each perceptual condition. This analysis (*Figure 4—figure supplement 3*) showed that for grayscale images, category information is significant in category-selective visual regions (bilateral LOC and FG), as well as parietal regions of the FPN and DMN (including bilateral parietal cortices of FPN, right Par Lat and PCC of DMN) (all p<0.05, FDR-corrected). For pre-disambiguation Mooney images, no significant category information was found in any ROI; however, for post-disambiguation Mooney images, significant category information was present in the right LOC and bilateral FG only (p<0.05, FDR-corrected).

Together, the above results suggest that FPN and DMN regions represent individual Mooney images more distinctly after disambiguation, and that this representation is more linked to individual image identity than image category.

## Disambiguation shifts neural representations toward the prior throughout the cortical hierarchy

We then examined how neural representation of a given Mooney image is altered by disambiguation, by comparing the neural activity pattern elicited by a Mooney image before or after disambiguation to that elicited by its matching gray-scale image. For intuitiveness, here we used representational similarity matrices (RSM), which correspond to 1 – RDM. The representational *similarity* between neural activity patterns elicited by a Mooney image and its matching gray-scale image constitutes an element along the diagonal of the 'Pre-Gray' square and the 'Post-Gray' square of the RSM (*Figure 3A*, bottom-right panel). We thus averaged the elements along each diagonal and compared their mean (red asterisk in *Figure 3A*, bottom-right).

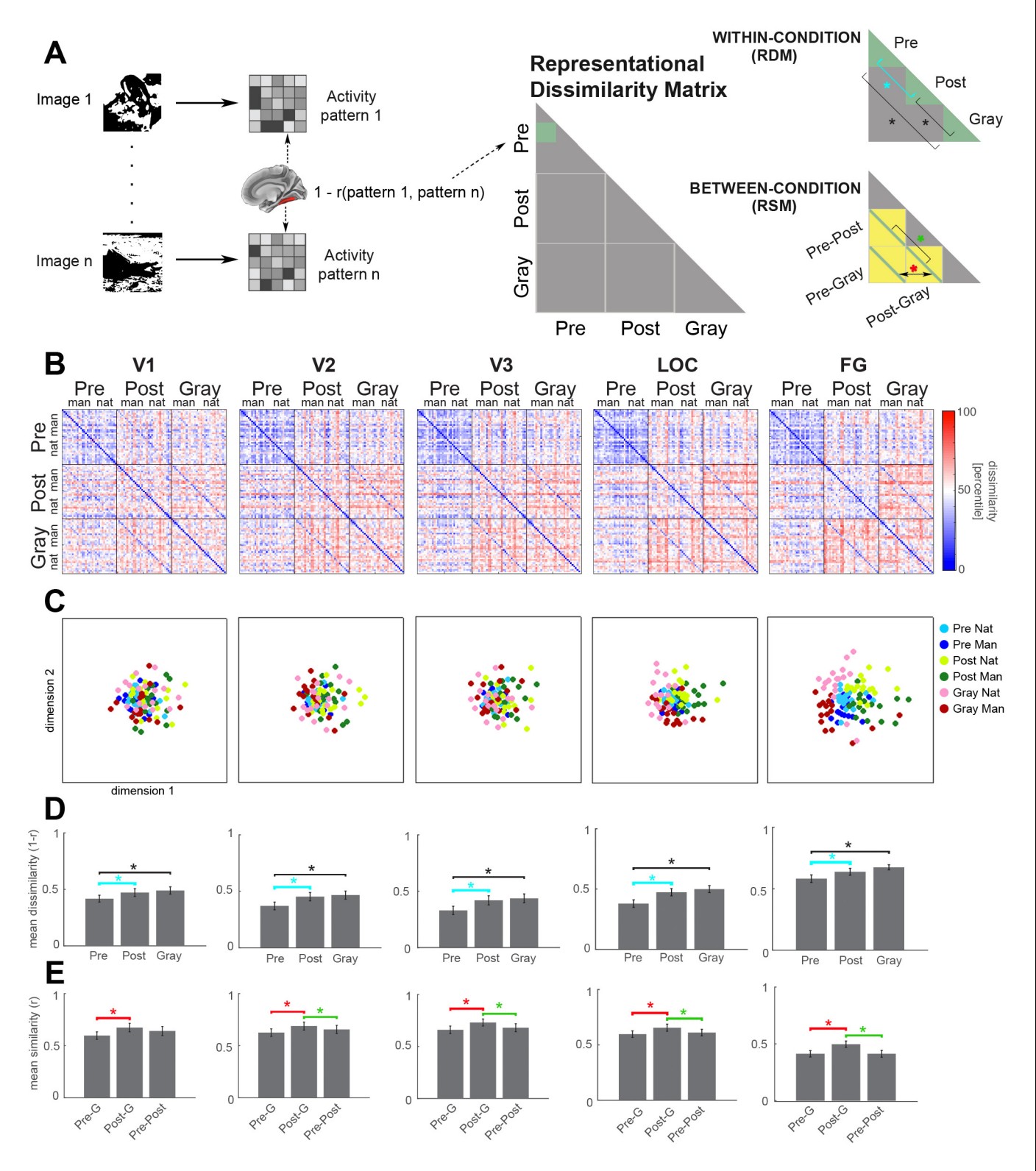

**Figure 3.** Neural representation format of individual images in visual regions. (**A**) Analysis schematic. Dissimilarity (1 – Pearson's r) between the neural response patterns to pairs of images was computed to construct the representational dissimilarity matrix (RDM) for each ROI. Two statistical analyses were performed: *First* (top-right panel, 'within-condition'), mean dissimilarity across image pairs was calculated for each condition (green triangles), and compared between conditions (brackets). Cyan bracket highlights the main effect of interest (disambiguation). *Second* (bottom-right panel, 'between-*Figure 3 continued on next page*

*Figure 3 continued*

condition'), the mean of between-condition diagonals (green lines) was compared. For ease of interpretation, this analysis was carried out on the representational similarity matrix (RSM). Each element in the diagonal represents the neural similarity between the same Mooney image presented in different stages (Pre-Post), or between a Mooney image and its corresponding gray-scale image (Pre-Gray and Post-Gray). (B) Group-average RDMs for V1, V2, V3, LOC and FG ROIs in the right hemisphere. Black lines delimit boundaries of each condition. Within each condition, natural ('nat') and man-made ('man') images are grouped together. (C) 2-D MDS plots corresponding to the RDMs in B. Pre-disambiguation, post-disambiguation, and gray-scale images are shown as blue, yellow-green, and pink-red dots, respectively. (D) Mean within-condition representational dissimilarity between different images for each ROI, corresponding to the 'within-condition' analysis depicted in A. (E) Mean between-condition similarity for the same or corresponding images for each ROI, corresponding to the 'between-condition' analysis depicted in A. In D and E, asterisks denote significant differences ($p < 0.05$, Wilcoxon signed-rank test, FDR-corrected), and error bars denote s.e.m. across subjects. Results from V4 are shown in *Figure 3—figure supplement 2*. Interactive 3-dimensional MDS plots corresponding to first-order RDMs for each ROI can be found at: https://gonzalezgarcia.github.io/mds.html.

DOI: https://doi.org/10.7554/eLife.36068.008

The following source data and figure supplements are available for figure 3:

**Source data 1.** RDM for each ROI in each subject. Includes source code to perform statistical analysis and produce *Figure 3 and 4*.
DOI: https://doi.org/10.7554/eLife.36068.011
**Figure supplement 1.** RSA results for left hemisphere visual regions.
DOI: https://doi.org/10.7554/eLife.36068.009
**Figure supplement 2.** RSA results for the right and left V4 (left and middle column, respectively), and the corresponding ROI size control analysis (right column).
DOI: https://doi.org/10.7554/eLife.36068.010

In all ROIs investigated across both hemispheres, from early and category-selective visual areas to FPN and DMN regions, the representational similarity between a post-disambiguation Mooney image and its matching gray-scale image is greater than that between a pre-disambiguation Mooney image and its matching gray-scale image (*Figure 3E and 4D*, red brackets; all $p < 0.05$, FDR-corrected, Wilcoxon signed-rank tests across subjects; and see panel D of *Figure 3—figure supplement 1*, *Figure 4—figure supplement 1*, *Figure 3—figure supplement 2*). Thus, disambiguation shifts neural representations of Mooney images significantly toward their priors throughout the cortical hierarchy.

To further examine the extent of this shift, we wondered whether the impact of priors could outweigh the influence of sensory input. In other words: Is a post-disambiguation Mooney image represented more similarly to the identical image presented in the pre-disambiguation period, or to the matching grayscale image, which, albeit physically different, elicits a similar perceptual recognition outcome (e.g., 'It's a crab!")? These alternatives can be adjudicated by comparing the diagonals of the 'Pre-Post' and 'Post-Gray' squares in the RSM (green asterisk in *Figure 3A*, bottom-right panel). Interestingly, in all ROIs except V1, post-disambiguation Mooney images were represented more similarly to their grayscale counterparts than to the same Mooney images shown before disambiguation (*Figure 3E and 4D*, green brackets; all $p < 0.05$, FDR-corrected, Wilcoxon signed-rank tests; and see panel D of *Figure 3—figure supplement 1*, *Figure 4—figure supplement 1*, *Figure 3—figure supplement 2*). A similar trend is present in V1, albeit not significant (right V1: $p = 0.06$; left V1: $p = 0.07$). This result suggests that across the cortical hierarchy, from V2 to FPN and DMN, content-specific neural activity patterns are shaped more strongly by prior and perceptual outcome than the immediate sensory input.

Since Gray-scale and post-disambiguation images are always presented later than Pre-disambiguation images, a potential confound to the results in *Figures 3* and *4* could be repetition suppression. To control for this, we repeated the above RSA analysis using two different sets of images: (1) images that were unrecognized in the pre-disambiguation stage and recognized in the post-disambiguation stage ('Disambiguation set'); and (2) images that were recognized in both the pre- and post-disambiguation stages ('Repetition set'). In both cases, for each participant, the same set of Mooney images were used in the Pre and Post conditions. Since repetition manipulation is identical between these two sets of images, this analysis allowed us to test whether disambiguation had an effect on neural activity above and beyond that of repetition. Using a repeated-measures ANOVA, we found a significant condition (pre- vs. post-disambiguation) × set ('disambiguation' vs. 'repetition') interaction for all neural effects of interest: higher between-image dissimilarity for post- than

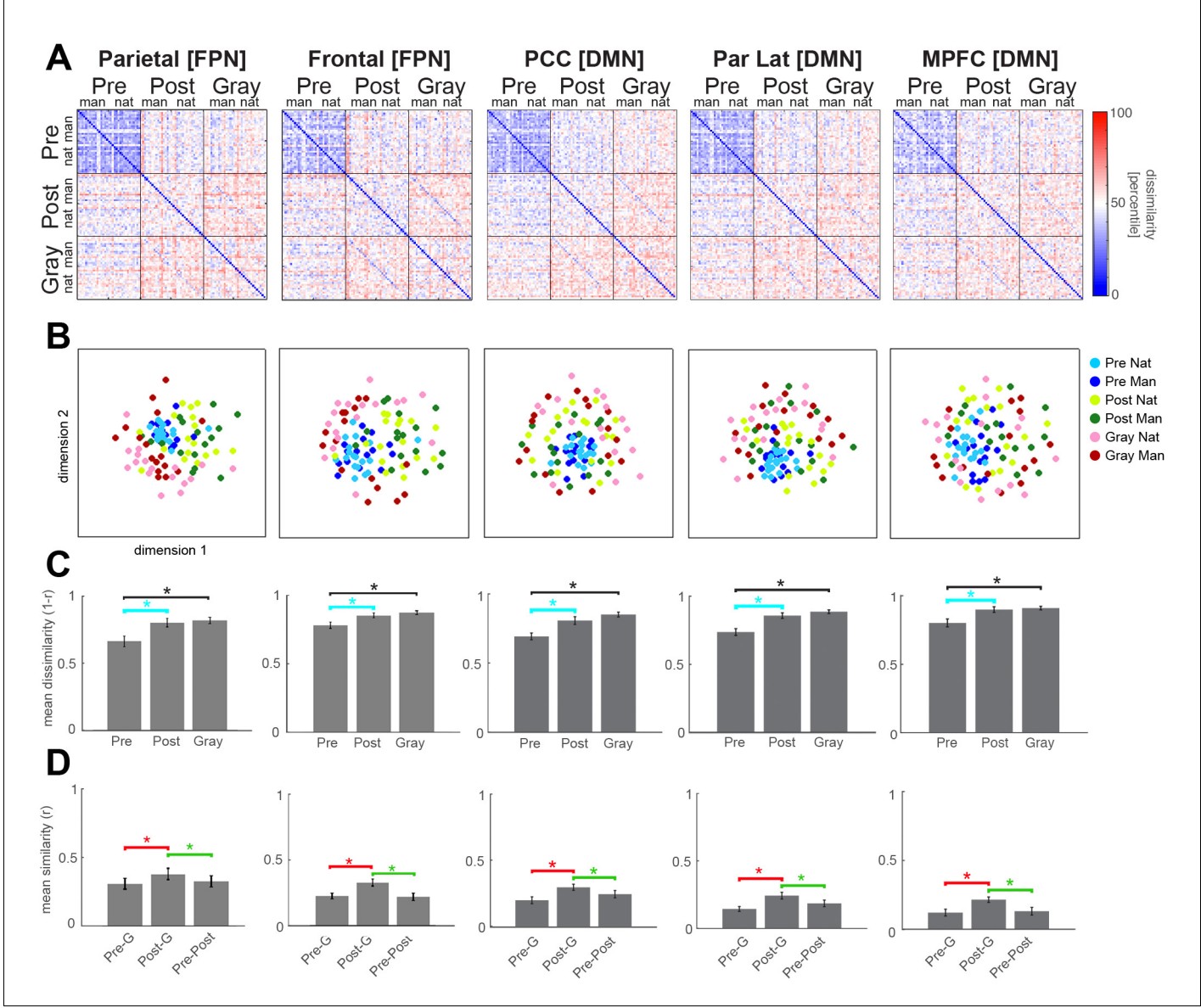

**Figure 4.** Neural representation format of individual images in frontoparietal regions. RSA results for FPN and DMN ROIs from the right hemisphere. Format is the same as in *Figure 3*. Interactive 3-dimensional MDS plots corresponding to first-order RDMs for each ROI can be found at: https://gonzalezgarcia.github.io/mds.html.

DOI: https://doi.org/10.7554/eLife.36068.012

The following figure supplements are available for figure 4:

**Figure supplement 1.** RSA results for left hemisphere FPN and DMN regions.

DOI: https://doi.org/10.7554/eLife.36068.013

**Figure supplement 2.** DMN results of the ROI size control analysis.

DOI: https://doi.org/10.7554/eLife.36068.014

**Figure supplement 3.** Image category (natural vs.manmade) information.

DOI: https://doi.org/10.7554/eLife.36068.015

pre-disambiguation images (cyan brackets in *Figure 3 and 4*, $F_{1,14}$ = 4.6, p=0.05); higher within-image similarity for post-Gray than pre-Gray comparison (red brackets, $F_{1,14}$ =14.9, p=0.002); higher within-image similarity for post-Gray than pre-post comparison (green brackets; $F_{1,14}$ = 6.3, p=0.02). Thus, this analysis revealed consistently larger neural effects in the Disambiguation set compare to

the Recognition set, demonstrating that disambiguation sculpted neural representations above and beyond the effects expected by mere repetition.

## Neural representation format across regions follow a principal gradient of macroscale cortical organization

The above results reveal extensive influences of priors on neural representations across the cortical hierarchy, and qualitatively similar effects of disambiguation across ROIs. But is there any systematic variation in neural representation format across brain regions? To address this, we carried out a second-order RSA (*Kriegeskorte et al., 2008*; *Guntupalli et al., 2016*). Second-order RSA calculates pairwise distance (1 - Spearman correlation) between the first-order RDMs of each ROI (*Figure 5A*, left, see Materials and methods for details). Each element of the resulting second-order RDM (*Figure 5B*) describes how similar two ROIs are in terms of their neural representation format. From *Figure 5B*, it can be seen that neural representational format is relatively similar between retinotopic visual areas, among LOC and fusiform regions, and across regions within the FPN or DMN, but relatively dissimilar between networks.

Like first-order RSA (*Figure 3 and 4*), the second-order RDM can be visualized using MDS to reveal relationships amongst ROIs in their neural representation format. The 2-D MDS solution of the second-order RDM reveals an organization in accordance with known anatomical and functional networks (*Figure 5C*; $r^2 = 0.92$, stress = 0.06, that is, the 2-D MDS solution explains 92% of variance in the high-dimensional RDM and has a high goodness-of-fit). This suggests that, consistent with the above impression, regions within a network have similar neural representation format. Moreover, regions are located along a gradient consistent with the visual hierarchy (Dimension 1), which ranges from V1 – V4, to LOC and FG, then frontoparietal regions. In addition, this hierarchy mirrors a principal gradient of macroscale cortical organization recently found based on resting-state functional connectivity and cortical anatomy, with sensory areas and DMN situated at the two extremes of the gradient, and FPN being intermediate (*Margulies et al., 2016*).

To assess whether the hierarchy of neural representation format is similar across perceptual conditions, we conducted a second-order RSA for each perceptual condition separately (*Figure 5A*, right). The 2-D MDS solutions for each within-condition second-order RDM are shown in *Figure 5D*, bottom, and all have high goodness-of-fit (pre-disambiguation: $r^2 = 0.91$, stress = 0.07; Gray-scale: $r^2 = 0.93$, stress = 0.06; post-disambiguation: $r^2 = 0.90$, stress = 0.09). Comparing the MDS solutions across perceptual conditions suggested that, indeed, there is a stable overall hierarchy of representation format across ROIs, despite changes in neural representation within each ROI due to disambiguation (as shown in *Figure 3 and 4*). An interesting, qualitative variation on this theme is that frontal regions of the FPN rise higher up in the hierarchy after disambiguation, such that the principal gradient (Dimension 1) separates frontal from parietal regions (*Figure 5D*, right panel), instead of DMN from FPN (*Figure 5D*, left and middle).

## Dimensionality of neural representation increases along the cortical hierarchy and following disambiguation

What might contribute to the stable, large-scale hierarchy of neural representation format? To shed light on this question, we estimated the dimensionality of the neural representational space for each ROI in each perceptual condition (i.e., the dimensionality of the within-condition 33 × 33 RDM shown in *Figure 3 and 4*; for details see Materials and methods, *RSA*), to test the hypothesis that more dimensions are needed to capture the complexity of the representational space in higher order regions. The group-averaged dimensionality for each ROI is shown in *Figure 6A–C*. A large-scale gradient can be seen in every perceptual condition: Dimensionality of neural representational space is relatively low in early visual areas and LOC, but substantially higher in FG, FPN and DMN regions.

To quantitatively compare the dimensionality of neural representational space across networks and perceptual conditions, we averaged dimensionality across ROIs for each network (while keeping LOC and FG separate), and conducted a two-way repeated-measures ANOVA (factors: network and perceptual condition; dependent variable: dimensionality; see *Figure 6D*). Both main effects were highly significant (network: $F_{56,4} = 41.96$, p=3.1e-16, $\eta^2_p = 0.75$; condition: $F_{28,2} = 22.27$, p=2e-6, $\eta^2_p = 0.61$), while the interaction was not significant (p=0.29, $\eta^2_p = 0.08$). Post-hoc tests (Bonferroni-

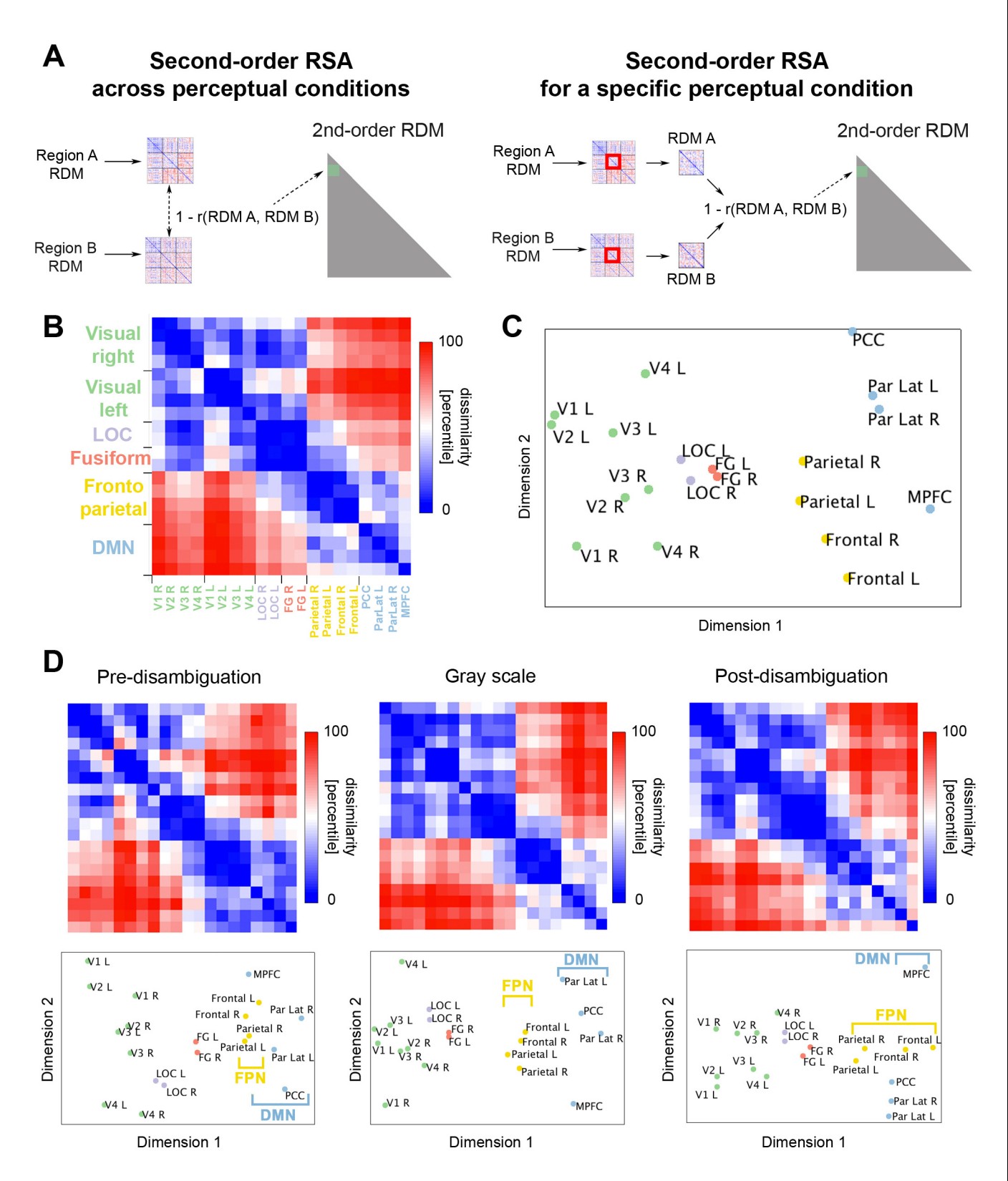

**Figure 5.** Relationship between neural representation format in different ROIs. (**A**) Analysis schematic for second-order RSA across perceptual conditions (left) and within each condition (right). The RDMs from different ROIs were averaged across subjects. Then, a dissimilarity value (1 –
*Figure 5 continued on next page*

*Figure 5 continued*

Spearman rho) was obtained for each pair of RDMs. (**B**) Across-condition second-order RDM, depicting the dissimilarities between first-order RDMs of different ROIs. (**C**) MDS plot corresponding to the second-order RDM show in B. (**D**) Second-order RDMs and corresponding MDS plots for the pre-disambiguation (left), gray-scale (middle), and post-disambiguation (right) conditions.

DOI: https://doi.org/10.7554/eLife.36068.016

The following source data is available for figure 5:

**Source data 1.** RDM for each ROI in each subject. Includes source code to perform second-order RSA and reproduce *Figure 5*.

DOI: https://doi.org/10.7554/eLife.36068.017

corrected) revealed that compared to early visual areas (2.9 ± 0.3; mean ±s.d. across subjects), dimensionality was significantly higher in FG (4.6 ± 0.9), FPN (4.6 ± 1) and DMN (4.3 ± 0.7; all *ps* < 0.001), but not LOC (3.1 ± 0.6; p>0.3). On the other hand, dimensionality did not differ significantly between FG and FPN or DMN (all *ps* > 0.08). Comparing across perceptual conditions, dimensionality was significantly higher for both post-disambiguation Mooney images (4 ± 1.1) and grayscale images (4.1 ± 1.1) than pre-disambiguation Mooney images (3.7 ± 0.7; both p<0.001), while the difference between post-disambiguation Mooney images and grayscale images was not significant (p>0.16).

Thus, these results reveal a stable hierarchy in the dimensionality of neural representational space: lowest in early visual areas, rising slightly in LOC, and highest in FG and frontoparietal areas. Moreover, in all these regions, dimensionality of neural representational space for Mooney images increased significantly following disambiguation, reaching roughly the same level as for grayscale images.

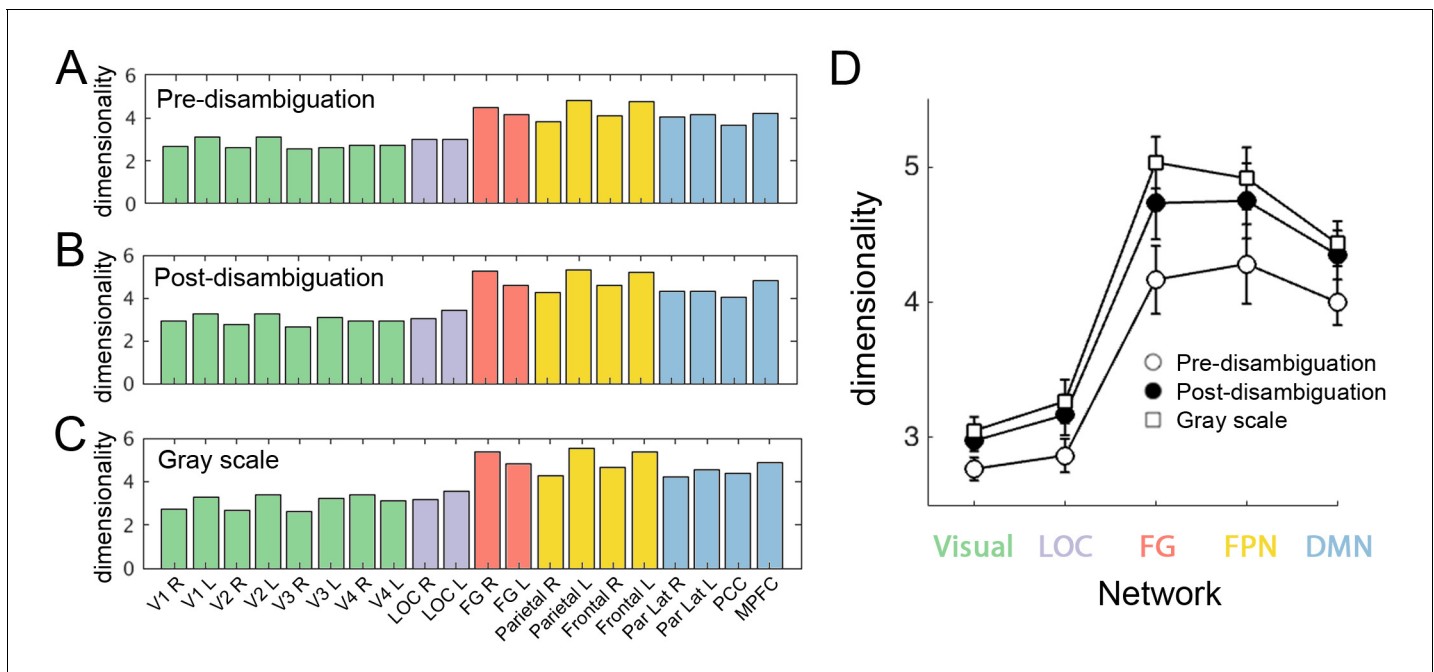

**Figure 6.** Dimensionality of neural representation space across ROIs and perceptual conditions. The dimensionality of neural representation (estimated by the number of dimensions needed in the MDS solution to achieve $r^2$ >0.9) for each ROI, in the pre-disambiguation (**A**), post-disambiguation (**B**), and gray-scale (**C**) condition, respectively. Each bar represents the mean dimensionality averaged across subjects for each ROI. (**D**) Group-averaged dimensionality for each network and condition. Error bars denote s.e.m. across subjects. Both the network (p=3.1e-16) and the condition (p=2e-6) factors significantly impacted dimensionality, while the interaction was not significant (p=0.29).

DOI: https://doi.org/10.7554/eLife.36068.018

The following source data is available for figure 6:

**Source data 1.** RDM for each ROI in each subject. Includes source code to perform dimensionality analysis and reproduce *Figure 6*.

DOI: https://doi.org/10.7554/eLife.36068.019

## Significant preservation of neural code following disambiguation in visual and parietal areas

Lastly, to quantitatively compare the impact of prior on neural representations across brain regions, we asked the following question for each region: How much of the neural code for pre-disambiguation Mooney images is preserved following disambiguation? To this end, we computed a Preservation Index (PI) by comparing the diagonal of the Pre-Post square of each ROI's RSM to the off-diagonal values (*Figure 7A*; for details see Materials and methods). The PI quantifies the similarity between the neural representation of a Mooney image in the pre- and post- disambiguation periods above and beyond its similarity to other images (i.e., is post-Image-A represented more similarly to pre-Image-A than to pre-Image-B?), and thus can be interpreted as cross-condition decoding of individual image identity (*Lee et al., 2012*). Therefore, a high PI suggests stronger preservation (as in 'better cross-decoding') of neural code before vs. after disambiguation, and a PI close to 0 suggests alteration of neural code (*Figure 7A*, right). For each subject, only Mooney images that were not-recognized in the pre-disambiguation period and recognized in the post-disambiguation period were used in this analysis (an additional analysis based on correct vs. incorrect verbal identification reports yielded very similar results).

This analysis revealed a large-scale trend of decreasing PI towards higher-order brain regions (*Figure 7B*). The neural code of Mooney images was significantly preserved following disambiguation in early visual areas, LOC, FG, and the FPN (all $p<0.05$, FDR-corrected, one-sample t-tests across subjects). However, the preservation index did not differ significantly from zero in DMN regions (all $p>0.05$). To assess differences in PI across networks, we performed a repeated-measures ANOVA with network (Visual, LOC, FG, FPN, and DMN) being the independent factor and PI being the dependent variable. This analysis revealed a significant effect of Network ($F_{56,4} = 41.86$, $p=3.27e-16$, $\eta^2_p = 0.75$). Post-hoc pairwise comparisons between individual networks were all significant (all $ps < 0.015$, Bonferroni corrected), except between Visual and LOC ($p=0.81$), or between FPN and DMN ($p=1$).

Further inspection of the data suggested differential PI in frontal and parietal regions of the FPN. To statistically assess this difference, we performed an additional repeated-measures ANOVA with only three levels for the Network factor (parietal regions of the FPN, frontal regions of the FPN, and the DMN). This analysis yielded again a significant effect of Network ($F_{36,2} = 6.86$, $p=0.003$, $\eta^2_p = 0.28$). Post-hoc comparisons (Bonferroni-corrected) revealed that PI was higher in FPN-parietal regions than FPN-frontal regions ($p=0.001$) or the DMN ($p=0.02$). In contrast, the difference between FPN-frontal regions and the DMN was not significant ($p=1$). While, by itself, these findings are consistent with the interpretation that higher-order regions such as the DMN and lateral prefrontal cortex may be unresponsive to the visual images, our earlier results rule out this possibility: neural representations in the DMN and frontal FPN regions are significantly impacted by disambiguation, becoming more distinct between individual images and shifting significantly toward the priors (*Figure 4*), with increased dimensionality in the neural representational space (*Figure 6*). Hence, neural representation is most profoundly altered by disambiguation in higher-order brain regions including the lateral prefrontal cortex and the DMN.

## Discussion

In summary, we observed extensive influences of prior experience on perceptual processing across the brain: Following the encoding of perceptual priors via disambiguation, Mooney images were represented more distinctly from each other, and more similarly to the prior-inducing grayscale image, throughout the cortical hierarchy – from early visual and category-selective areas to FPN and DMN. These results reveal, unambiguously, content-specific neural representations during prior-guided visual processing in the DMN and FPN. Interestingly, despite the prior's pervasive influence on neural activity, which was more pronounced in higher-order brain regions, the neural representation format across brain regions followed a stable macroscale hierarchy that mirrors one recently uncovered in resting-state functional connectivity, with visual areas and DMN situated at the two extremes, and FPN being intermediate (*Margulies et al., 2016*). Along this hierarchy, the dimensionality (i.e., complexity) of neural representational space increases towards higher-order brain areas, and rises in all regions after the establishment of perceptual priors.

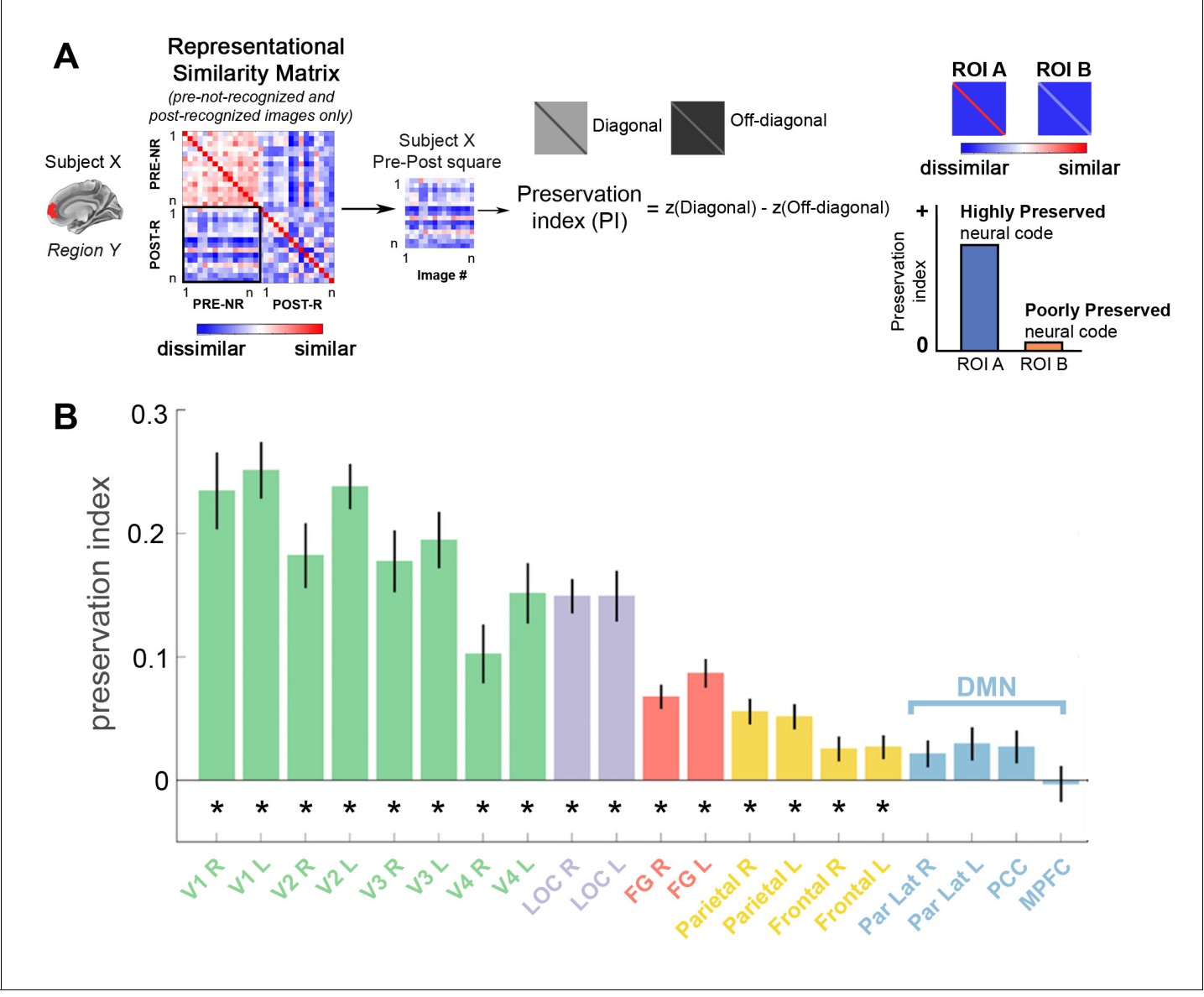

**Figure 7.** Significant preservation of neural code is found in visual and FPN regions. (**A**) Analysis schematic. For each ROI in each subject, a representational similarity matrix (RSM) is constructed using the set of Mooney images that are not recognized in the pre-disambiguation period (PRE-NR) and recognized in the post-disambiguation period (POST-R). The Pre-Post square was extracted from this RSM, and the r-values were Fisher-z-transformed. Then, the difference between diagonal and off-diagonal elements was calculated for each subject, termed 'Preservation Index' (PI). A significant positive PI means that a post-disambiguation Mooney image is represented more similarly to the same image than other images shown in the pre-disambiguation period. (**B**) Group-averaged PI values for each ROI. Error bars denote s.e.m. across subjects. Asterisks denote significant PI values (p<0.05, FDR-corrected, one-sample t-test across subjects).

DOI: https://doi.org/10.7554/eLife.36068.020

The following source data is available for figure 7:

**Source data 1.** RSM (as shown in *Figure 7A*, left) for each ROI in each subject. Includes source code to perform analysis and reproduce *Figure 7*.
DOI: https://doi.org/10.7554/eLife.36068.021

Previous studies have shown that the DMN is active in tasks that rely on information retrieved from memory, such as spontaneous thoughts, autobiographical and semantic memory, planning about the future, and executive control tasks requiring long-term or working memory (e.g., [*Buckner et al., 2008*; *Christoff et al., 2009*; *Shapira-Lichter et al., 2013*; *Konishi et al., 2015*; *Spreng and Grady, 2010*; *Spreng et al., 2014*]). Yet, whether the DMN may be involved in

representing prior information that impacts *perceptual* processing has heretofore remained unknown. Our results herein reveal content-specific neural activity in the DMN during prior-guided visual processing for the first time. Specifically, our results demonstrated that: (*i*) image-specific information increases following disambiguation in DMN regions; (*ii*) neural representations of disambiguated Mooney images shift significantly toward their priors; (*iii*) individual images are represented by DMN regions in a higher-dimensional space following disambiguation. Importantly, the enhanced image-level information in DMN regions following disambiguation could not be attributed to an overall higher level of signal, since the activity is actually closer to baseline than that elicited by pre-disambiguation Mooney images. These results suggest that, even in an externally oriented perceptual task, DMN regions cannot merely encode nonspecific effects such as arousal (*Gorlin et al., 2012*) or task difficulty (*Singh and Fawcett, 2008*) as previously postulated. Instead, they are actively involved in the processing of these visual images, as their representation of a given visual stimulus was altered by the presence of a perceptual prior and shifted significantly toward that prior. The profound changes in DMN representations following fast, automatic disambiguation uncovered herein resonate with a previous finding showing that slow, deliberate perceptual training alters resting-state functional connectivity between DMN regions and the visual cortex (*Lewis et al., 2009*). Our results are also consistent with a recent study showing greater DMN activation to objects than color patches during a working memory task (*Murphy et al., 2018*), which, speculatively, may reflect stronger priors for the former learnt through past experiences, although differential stimulus complexity might provide another account for this previous finding.

Consistent with our prior hypothesis, the FPN exhibited an intermediate pattern compared to DMN and visual areas in this task. Although all these regions exhibited enhanced image-level information, a higher-dimensional neural representational space, and activity patterns that shifted towards the prior-inducing images following disambiguation, using a second-order RSA, we established that the FPN's neural representational format was indeed intermediate between that of visual areas and DMN. In addition, the extent to which priors altered neural representations in the FPN was also intermediate of that in the visual areas and DMN (*Figure 7*). Because all disambiguated Mooney images required the same response (counterbalanced button presses to indicate 'recognized'), these findings in the FPN do not fit easily with a previous suggestion that the FPN only encode the distinction between targets and non-targets (*Erez and Duncan, 2015*). Instead, they point to content-specific representations relevant to *perceptual* processing in the FPN. Interestingly, although the global hierarchy of neural representation format across cortical areas remained stable across perceptual conditions, following disambiguation frontal areas of the FPN moved up the hierarchy, suggesting that they may have a special role in utilizing priors to guide perceptual processing. These results amplify limited existing evidence for frontal areas' involvement in prior-guided visual perception (*Wang et al., 2013*; *Bar et al., 2006*; *Imamoglu et al., 2012*; *Summerfield et al., 2006*).

Outside the DMN and FPN, our findings support earlier reports of increased image-level information in early and category-selective visual regions following disambiguation, as well as a shift of neural representation in visual areas toward the prior-inducing image (*Hsieh et al., 2010*; *Gorlin et al., 2012*; *van Loon et al., 2016*). An unresolved issue in these previous studies was whether image-specific information increases in V1 following disambiguation (*Gorlin et al., 2012*; *van Loon et al., 2016*); our results suggest that this is indeed the case, in line with the finding in (*van Loon et al., 2016*). In addition, an intriguing, novel finding in visual areas in the current study concerns the difference in neural representation between the LOC and the FG. Although both are high-level, category-selective visual areas, and both are consistently placed between early visual areas (V1 – V4) and FPN in the macroscale cortical hierarchy of neural representational format (*Figure 5*), several findings converge to show a clear hierarchy between them, with the FG being a higher-order region than LOC: First, FG is consistently placed higher up than LOC along the cortical hierarchy across perceptual conditions (*Figure 5*); second, FG consistently exhibits higher dimensionality in its neural representational space than LOC across perceptual conditions; last, disambiguation alters neural representation more strongly in FG than LOC. Both higher dimensionality and larger impact of prior are features associated with higher-order brain areas (*Figure 6 and 7*). These results revealing a hierarchy between LOC and FG are compatible with the FG being located more anteriorly than the LOC (*Figure 2—figure supplement 3*) and previous report of a representational gradient between these two regions (*Haushofer et al., 2008*).

The Mooney image disambiguation effect is a dramatic example of one-shot learning, where a single exposure to an event leaves a long-term memory in the brain that influences future perception of a different, but related stimulus. While the poster child of one-shot memory is episodic memory, the perceptual prior acquired during Mooney image disambiguation differs from episodic memory by being non-self-referential and potentially non-declarative (*Ludmer et al., 2011*; *Chang et al., 2016*). The presence of this perceptual prior is probed by whether the subject recognizes the corresponding Mooney image, not by whether the subject recalls having seen either the Mooney or Gray-scale image earlier (the episodic memory of the event itself). Supporting this distinction, a previous study found that activity in mid-level visual regions, MPFC, and amygdala, but not hippocampus, during gray-scale image viewing predicted successful disambiguation of Mooney images (*Ludmer et al., 2011*). In accordance, we did not find a significant increase in image-specific information following disambiguation in anatomically defined hippocampal ROIs. Compared to episodic memory, the neural mechanisms underlying experience-induced perceptual priors have been much less studied. Understanding how a single experience may leave a prior in the brain that alters future perception addresses a fundamental scientific question with important clinical implications. For instance, if prior experiences overwhelm sensory input, hallucinations may follow, as happens in post-traumatic stress disorder (*Clancy et al., 2017*). Similarly, psychosis-prone individuals have a larger disambiguation effect than control subjects in the Mooney image paradigm, consistent with the idea that internal priors play an especially strong role in shaping perception during psychosis (*Teufel et al., 2015*).

In conclusion, we observed that prior experience impacts visual perceptual processing throughout the cortical hierarchy, from occipitotemporal visual regions to frontoparietal and default-mode networks. In addition, we observed a macroscale hierarchy of neural representation format, with increased dimensionality in neural representational space and more profound influences of perceptual prior on neural representations in higher-order frontoparietal regions. These results reveal content-specific neural representations in frontoparietal and default-mode networks involved in prior-guided visual perception.

## Materials and methods

### Subjects

Twenty-three healthy volunteers participated in the study. All participants were right-handed and neurologically healthy, with normal or corrected-to-normal vision. The experiment was approved by the Institutional Review Board of the National Institute of Neurological Disorders and Stroke. All subjects provided written informed consent. Four subjects were excluded due to excessive movements in the scanner, leaving 19 subjects for the analyses reported herein (age range = 19–32; mean age = 24.6; 11 females).

### Visual stimuli

The construction and selection of Mooney images and corresponding gray-scale images were described in detail in a previous study (*Chang et al., 2016*). Briefly, Mooney and gray-scale images were generated from gray-scale photographs of real-world man-made objects and animals selected from the Caltech (http://www.vision.caltech.edu/Image_Datasets/Caltech101/Caltech101.html) and Pascal VOC (http://host.robots.ox.ac.uk/pascal/VOC/voc2012/index.html) databases. First, gray-scale images were constructed by cropping gray-scale photographs with a single man-made object or animal in a natural setting to 500 × 500 pixels and applying a box filter. Mooney images were subsequently generated by thresholding the gray-scale image. Threshold level and filter size were initially set at the median intensity of each image and 10 × 10 pixels, respectively. Each parameter was then titrated so that the Mooney image was difficult to recognize without first seeing the corresponding gray-scale image. Out of an original set of 252 images, 33 (16 were man-made objects, the rest animals – unbeknownst to the subjects) were chosen for the experiment via an initial screening procedure, which was performed by six additional subjects recruited separately from the main experiment. Images with the largest disambiguation effect – assessed by changes in difficulty rating before vs. after disambiguation – were chosen for this study. Images were projected onto a screen

located at the back of the scanner and subtended approximately 11.9 × 11.9 degrees of visual angle.

## Task paradigm

Each trial started with a red fixation cross presented in the center of the screen for 2 s, and then a Mooney image or a gray-scale image presented for 4 s. The fixation cross was visible during image presentation, and subjects were instructed to maintain fixation throughout. A 2 s blank period appeared next, followed by a brighter fixation cross (lasting 2 s) that prompted participants to respond (see *Figure 1A*). Participants were instructed to respond to the question 'Can you recognize and name the object in the image?' with an fMRI-compatible button box using their right thumb. Trials were grouped into blocks, using a design similar to previous studies (*Gorlin et al., 2012*; *Chang et al., 2016*). Each block contained fifteen trials: three gray-scale images followed by six Mooney-images and a shuffled repetition of the same six Mooney-images. Three of the Mooney images had been presented in the previous run and corresponded to the gray-scale images in the same block (these are post-disambiguation Mooney images). The other three Mooney-images were novel and did not match the gray-scale images (pre-disambiguation); their corresponding gray-scale images would be presented in the following run. Each fMRI run included three blocks of the same images; within each block, image order was shuffled but maintained the same block structure (gray-scale followed by Mooney images). After each run, a verbal test was conducted between fMRI runs. During the verbal test, the six Mooney images from the previous run were presented one by one for 4 s each, and participants were instructed to verbally report the identity of the image. Each participant completed 12 runs. In order for all Mooney images to be presented pre- and post-disambiguation, the first and last runs were 'half runs'. The first run contained only three novel Mooney images (pre-disambiguation). The last run consisted of 3 gray-scale images and their corresponding post-disambiguation Mooney images. The total duration of the task was ~90 min. The order of Mooney images presentation was randomized across participants.

## Data acquisition and preprocessing

Imaging was performed on a Siemens 7T MRI scanner equipped with a 32-channel head coil (Nova Medical, Wilmington, MA, USA). T1-weighted anatomical images were obtained using a magnetization-prepared rapid-acquisition gradient echo (MP-RAGE) sequence (sagittal orientation, 1 × 1×1 mm resolution). Additionally, a proton-density (PD) sequence was used to obtain PD-weighted images also with 1 × 1 × 1 mm resolution, to help correct for field inhomogeneity in the MP-RAGE images (54). Functional images were obtained using a single-shot echo planar imaging (EPI) sequence (TR = 2000 ms, TE = 25 ms, flip angle = 50°, 52 oblique slices, slice thickness = 2 mm, spacing = 0 mm, in-plane resolution = 1.8×1.8 mm, FOV = 192 mm, acceleration factor/ GRAPPA = 3). The functional data were later resampled to 2 mm isotropic voxels. Respiration and cardiac data were collected using a breathing belt and a pulse oximeter, respectively. The breathing belt was wrapped around the upper abdomen, and the pulse oximeter was placed around the left index finger. Physiological data were collected simultaneously with fMRI data using the AcqKnowledge software (Biopac Systems, Inc.).

For anatomical data preprocessing, MP-RAGE and PD images were first skull-stripped. Then, the PD image was smoothed using a 2 mm full-width at half maximum (FWHM) kernel. Afterwards, the MP-RAGE image was divided by the smoothed PD image to correct for field inhomogeneity.

Functional data preprocessing started with the removal of physiological (respiration- and cardiac-related) noise using the RETROICOR method (*Glover et al., 2000*). The next preprocessing steps were performed using the FSL package (http://fsl.fmrib.ox.ac.uk/fsl/fslwiki/FSL). These included: ICA cleaning to remove components corresponding to physiological or movement-related noise,>0.007 Hz high-pass filtering (corresponding to a temporal period of 150 s) to remove low-frequency drifts, rigid-body transformation to correct for head motion within and across runs, slice timing correction to compensate for systematic differences in the time of slice acquisition, and spatial smoothing with a 3 mm FWHM Gaussian kernel. Last, images were registered to the atlas in two steps. First, functional images were registered to the individual's anatomical image via global rescale (7 degrees of freedom) transformations. Affine (12 degrees of freedom) transformations were then used to register the resulting functional image to a 2 × 2 × 2 mm MNI atlas. Registration to MNI space was

performed on voxel-wise GLM and searchlight MVPA results to obtain population-level whole-brain maps. RSA analyses were conducted in individual subject's functional data space and results were pooled across subjects.

## Lateral occipital complex (LOC) functional localizer

133 images of objects (including fruits, vegetables, and man-made objects) were extracted from the Bank of Standardized Stimuli (BOSS - https://sites.google.com/site/bosstimuli). From each object image, a corresponding image of phase-shuffled noise was created. Each trial consisted of a brief blank period of 100 ms, followed by an image or shuffled noise, presented for 700 ms. Each block contained 16 trials, and subjects were instructed to maintain fixation on a dimmed red cross at the centre of the screen throughout the block. Within each block, four trials consisted of repetitions of the previous item. Subjects were instructed to press a button when the same image of object or shuffled noise appeared twice in a row, which occurred randomly within the block. The single localizer run (lasting ~3 min) contained 10 blocks, 5 of images and 5 of shuffled noise, each of which started with a warning cross lasting 4 s. The LOC functional localizer data were analyzed using a GLM, which included regressors for image and shuffled-noise events, as well as an additional regressor that modelled the warning cross at the beginning of each block. A significance map of the Image > Noise contrast was obtained for each subject.

## Retinotopy functional localizer

Subjects were shown a circular checkerboard of 100% contrast and 21° diameter through a bar aperture that progressed through the screen to cover the entire visual field. One sweep included 18 steps, one every TR (2 s), taking a total of 36 s. During the single retinotopy scan, eight sweeps were performed, accounting for four different orientations (left, right, bottom, up) and two directions. The specific pattern of sweeps was as follows: left-right, bottom right-top left, top-down, bottom left-top right, right-left, top left-bottom right, bottom-top, and top right-bottom left. Participants were instructed to stare at a fixation cross in the center of the screen and press a button when it changed from green to red. These changes occurred in a semi-random fashion, with approximately two changes per sweep. The total duration of the retinotopy scan was ~5 min.

Retinotopy data preprocessing was performed using custom-written AFNI code. Preprocessing included motion correction to the first volume in the run, detrending and removal of linear trends (such as scanner drift) using linear least-squares, and spatial smoothing (5 mm FWHM kernel). Population receptive fields (pRF) (*Dumoulin and Wandell, 2008*) analysis was performed using an AFNI-based implementation (*Silson et al., 2015*; *Silson et al., 2016*). Briefly, for each voxel, two algorithms found the best fit (in terms of location in the field of view, and size of the receptive field) between the predicted and observed time-series, by minimizing the least-squares error between these two. The output of the model included X and Y location, sigma (size) and $R^2$ estimate of the best fitting model. In order to delineate the different visual regions, polar angle and eccentricity components were created from X and Y data. These components were projected onto 3-D hemispherical surface reconstruction of individual subject's gray and white matter boundaries (obtained using the recon-all command in Freesurfer, http://freesurfer.net/). Following the criteria described in (*Silson et al., 2016*), we defined the following field maps in left and right hemispheres of each subject: V1, V2d, V2v, V3d, V3v and V4. Dorsal and ventral portions of V2 and V3 were merged into one single ROI for each hemisphere, respectively.

## General Lineal Model (GLM) analysis

A GLM was used to assess changes in activation magnitude across conditions. At the individual subject level, a model was constructed including regressors for pre-disambiguation not-recognized, pre-disambiguation recognized, post-disambiguation recognized Mooney images, and gray-scale images. Two different versions of the model were created, one based on subjective recognition responses and the other based on verbal identification responses. In the former, an image was defined as not-recognized if the subject responded 'yes' in two or fewer out of the 6 presentations of that image; it was defined as recognized if the subject responded 'yes' in four or more presentations of the image (see *Figure 1D*). Importantly, the same set of images are included in the 'pre-disambiguation not-recognized' and 'post-disambiguation recognized' conditions.

All regressors were convolved with a hemodynamic response function (HRF) following a Gamma shape (half-width of 3 s and lag of 6 s). In addition, temporal derivatives of each regressor were also included in the model. At the population level, parameter estimates of each regressor were entered into a mixed-effects analysis to obtain group-level estimates (FLAME1 estimation method). To correct for multiple comparisons, a cluster-defining threshold of p<0.001 (z > 3.1) and minimum cluster size of 17 voxels was applied (corresponding to p<0.05, FWE-corrected).

## Multivariate pattern analysis (MVPA)

MVPA was performed using The Decoding Toolbox (*Hebart et al., 2014*) and custom-written codes in MATLAB. First, in order to obtain a beta estimate for each individual image and subject, a new GLM was fit to the preprocessed fMRI data in the native space. For each subject, each regressor included all presentations of a given image in a given condition.

To decode the status of the image (pre-disambiguation not-recognized from post- disambiguation recognized), a searchlight analysis across the whole brain was conducted using 6-voxel radius spheres and following an n-fold cross-validation scheme. As in the GLM analysis, the same set of images were included in both conditions. In each fold, all samples but two (one from each class) were used to train the classifier (linear support vector machine (SVM); cost parameter = 1) which was then tested on the remaining two samples. The accuracy value was averaged across folds and assigned to the center voxel of each sphere. To assess significance at the population level, a non-parametric permutation-based approach was used, with 5000 shuffles. A threshold-free cluster enhancement algorithm (*Smith and Nichols, 2009*) was then used to find significant clusters (p<0.01, FWE-corrected) on the resulting map.

## Region-of-interest (ROI) definition

Individually defined ROIs of DMN [medial prefrontal cortex (MPFC), posterior cingulate cortex (PCC), left and right lateral parietal cortex (Par Lat)] were extracted from each subject's estimate of the post-recognized > pre not-recognized GLM contrast in the native space (*Figure 2A*). The same approach was applied to the LOC localizer data to obtain LOC ROIs for each subject, using the 'Image > Shuffled Noise' contrast. To define FPN ROIs, the corrected, population-level statistical map of the pre-not-recognized vs. post-recognized decoding analysis (*Figure 2C*) was used to define frontal and parietal clusters, bilaterally. Frontal clusters of the decoding results roughly matched the frontal areas of FPN reported in previous studies (*Power et al., 2011*) (*Figure 2—figure supplement 4C*) and were used to define our FPN frontal ROIs (*Figure 2—figure supplement 4B*, red). In contrast, parietal clusters of the decoding result encompassed the parietal regions of both FPN and DMN (compare panels A and C in *Figure 2—figure supplement 4*). Thus, we selected those voxels of our decoding map overlapping with previously reported parietal regions of the FPN (*Power et al., 2011*) to define our FPN parietal ROIs (*Figure 2—figure supplement 4B*, red, also see *Figure 2—figure supplement 3*). These population-level ROIs were then registered back to the native space of each subject to obtain individual ROIs. Additionally, we obtained left and right fusiform gyrus ROIs by extracting the template regions from the Harvard-Oxford Cortical Structural Atlas (https://fsl.fmrib.ox.ac.uk/fsl/fslwiki/Atlases) and registering these back to the native space of each subject. Last, V1-V4 were defined in each subject's native space based on the retinotopic localizer described earlier.

Finally, since DMN and FPN ROIs were defined using the same data set, albeit by independent analyses (GLM and MVPA, respectively) from the ROI-based RSA analyses, to exclude any concern about potential influences of double dipping, we repeated the RSA analyses using FPN and DMN ROIs defined from an independent resting-state data set (*Power et al., 2011*). The results obtained using these ROIs replicated those reported in *Figures 4–7*.

## Representational similarity analysis (RSA)

We performed ROI-based RSA on the activity patterns of individual images to assess the impact of disambiguation on their neural representation. These activity patterns were derived from GLM's beta weights as described above in the section MVPA. For each subject and ROI, we calculated the representational distance (1 – Pearson's r) using activity pattern across voxels between pairs of images from all conditions (33 pre-disambiguation, 33 post-disambiguation and 33 gray-scale),

obtaining a 99 × 99 symmetrical representational dissimilarity matrix (RDM) (*Figure 3A*). The diagonal elements of the RDM have value 0 and are not used in further analyses. In order to conduct group-level statistics, all images from all subjects were used and this analysis thus did not depend on subjective recognition or verbal identification responses.

To facilitate visualization, the RDM was projected to a 2-dimensional plot using non-metric multidimensional scaling (MDS; criterion = stress1), where each dot corresponds to the neural representation of one image, and distances between dots preserve the original representational distances in the RDM as much as possible.

Based on each ROI's RDM, two tests were conducted. First, we assessed whether individual images are represented more distinctly in one perceptual condition than another, by comparing the mean of the lower triangle of within-category squares of the RDM (*Figure 3A*, top-right; for example, comparing the mean representational distance between pre-disambiguation Mooney images to the mean distance between post-disambiguation images). Second, we investigated the representational *similarity* of an identical or related image presented in different conditions (e.g., comparing the representational similarity between the same image presented pre and post disambiguation to that between a post-disambiguation Mooney image and its corresponding gray-scale image), taking the diagonals of between-condition squares of the representational similarity matrix (RSM, where each element includes the Pearson's r-value between two images; *Figure 3A*, bottom-right). In both cases, differences between conditions were assessed by paired Wilcoxon signed-rank tests across subjects, and results were FDR-corrected for multiple comparisons.

To quantify the amount of category information in each ROI under each perceptual condition, for each subject, we first extracted the within-condition (Pre-Pre; Post-Post; Gray-Gray) portions of the RDM (*Figure 4—figure supplement 3A*). Within each condition, images are sorted based on category: Natural (Nat) and Manmade (Man). We then computed the mean dissimilarity across image pairs where both images are of the same category but are non-identical (i.e., off-diagonal elements of the Nat-Nat square and Man-Man square), as well as the mean dissimilarity across image pairs where the two images are of different categories (i.e., elements in the Nat-Man square). The difference between the two mean dissimilarity scores was computed for each subject, and subjected to a group-level Wilcoxon signed-rank test against 0. A significant difference, with lower within-category dissimilarity than between-category dissimilarity, suggests significant category-level information.

Finally, to estimate the dimensionality of neural representational space for each ROI in each perceptual condition, we performed MDS using progressively increasing numbers (from 2 to 10) of dimensions. For each MDS solution, we assessed its goodness-of-fit using $r^2$, which indexes the proportion of variance in the original RDM explained by the MDS. To obtain $r^2$, we first computed the pairwise Euclidean distances of the coordinates resulting from the MDS, and then calculated the squared correlation between this new RDM (corresponding to the N-dimensional MDS solution) with the original RDM. For each ROI and perceptual condition, we then determined the smallest number of dimensions needed for the MDS to achieve a high goodness-of-fit (set at a threshold of $r^2 > 0.90$).

## Second-order RSA

To perform second-order RSA (*Kriegeskorte et al., 2008*; *Guntupalli et al., 2016*), we calculated the distance (using 1 - Spearman rho) of group-averaged RDM between every pair of ROIs (*Figure 5A*). As in first-order RSA, we applied MDS to obtain a 2-D plot of the representational distances amongst ROIs (*Figure 5C*), such that distance between any two ROIs in this plot is inversely proportional to how similarly they represent the set of images used in this experiment. The goodness-of-fit of the 2-D MDS solution was assessed using two metrics: $r^2$, which quantifies the fraction of variance in the original high-dimensional RDM captured by the 2-D MDS solution; and stress, where a stress < 0.1 indicates high goodness-of-fit. This analysis was first calculated across perceptual conditions (*Figure 5A*, left), that is, using the entire group-averaged RDMs. The analysis was then repeated for each condition separately, using the within-condition square of the group-averaged RDMs (*Figure 5A*, right).

## Preservation Index (PI) for investigating changes in neural code due to disambiguation

To avoid potential confounds (see Results), this analysis was performed in new RDMs computed exclusively with pre-not-recognized and post-recognized images. Based on the representational similarity matrix (RSM) from each ROI, we used the following formula to compute a preservation index by comparing the diagonal and off-diagonal elements in the pre-post square of the matrix (*Figure 7A*):

Preservation index = mean(z(diagonal similarity)) – mean(z(off-diagonal similarity)), where z denotes Fisher's z-transform, which transforms Pearson's r-values into a normal distribution. We then performed a one-sample t-test against 0 on the PI across subjects for each ROI, and the results were FDR-corrected for multiple comparisons. A significant positive PI suggests that the neural representation of a post-disambiguation Mooney image is significantly more similar to the same image than to other images presented in the pre-disambiguation period.

An additional analysis was carried out using a normalized PI metric as follows:

$$\text{Preservation index (normalized)} = \frac{(\text{mean(z(diagonal similarity))} - \text{mean(z(off} - \text{diagonal similarity)))})}{(\text{mean(z(off} - \text{diagonal similarity)))}}.$$

A one-sample Wilcoxon signed-rank test (because the ratio of two normally distributed variables no longer follows a normal distribution) on the normalized PI across subjects yielded an identical pattern of results as the above analysis.

## Source data and code

Data and Matlab code used for analyses and producing results in the main figures are available as Source Data and Source Code with this article.

## Acknowledgements

This research was supported by the Intramural Research Program of the National Institutes of Health/National Institute of Neurological Disorders and Stroke, New York University Langone Medical Center, Leon Levy Foundation and Klingenstein-Simons Fellowship (to BJH). CGG was supported by the Department of State Fulbright program. We thank Edward Silson for help with implementing population receptive field retinotopic mapping, Souheil Inati and Vinai Roopchansingh for help with implementing scanner pulse sequence, and Catie Chang for help with using RETROICOR.

## Additional information

### Funding

| Funder | Grant reference number | Author |
|---|---|---|
| Leon Levy Foundation | Leon Levy Neuroscience Fellowship | Biyu J He |
| Esther A. and Joseph Klingenstein Fund | Klingenstein-Simons Neuroscience Fellowship | Biyu J He |
| Fulbright Association | The Fulbright Program | Carlos González-García |
| National Institute of Neurological Disorders and Stroke | Intramural Research Program | Biyu J He |

The funders had no role in study design, data collection and interpretation, or the decision to submit the work for publication.

### Author contributions

Carlos González-García, Data curation, Software, Formal analysis, Funding acquisition, Validation, Investigation, Visualization, Methodology, Writing—original draft, Writing—review and editing; Matthew W Flounders, Data curation, Investigation; Raymond Chang, Conceptualization, Data curation, Formal analysis, Investigation, Visualization, Methodology; Alexis T Baria, Conceptualization,

Methodology; Biyu J He, Conceptualization, Resources, Data curation, Supervision, Funding acquisition, Validation, Methodology, Writing—original draft, Project administration, Writing—review and editing

### Author ORCIDs
Carlos González-García http://orcid.org/0000-0001-6627-5777
Biyu J He http://orcid.org/0000-0003-1549-1351

### Ethics
Human subjects: Twenty-three healthy volunteers participated in the study. The experiment was approved by the Institutional Review Board of the National Institute of Neurological Disorders and Stroke. All subjects provided written informed consent.

### Decision letter and Author response
Decision letter https://doi.org/10.7554/eLife.36068.029
Author response https://doi.org/10.7554/eLife.36068.030

## Additional files

### Supplementary files
• Transparent reporting form
DOI: https://doi.org/10.7554/eLife.36068.022

### Data availability
All data generated or analysed during this study are included in the manuscript and supporting files. Source data files have been provided for Figures 3–7.

The following previously published datasets were used:

| Author(s) | Year | Dataset title | Dataset URL | Database, license, and accessibility information |
|---|---|---|---|---|
| Fei-Fei L, Fergus R, Perona P | 2007 | Caltech 101 dataset | http://www.vision.caltech.edu/Image_Datasets/Caltech101/Caltech101.html | Publicly available |
| Everingham M, Van Gool L, Williams CK, Winn J, Zisserman A | 2010 | Pascal VOC database | http://host.robots.ox.ac.uk/pascal/VOC/voc2012/index.html | Publicly available |

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
