## [Decision Letter]

Thank you for submitting your article "Content-specific activity in frontoparietal and default-mode networks during prior-guided visual perception" for consideration by *eLife*. Your article has been reviewed by three peer reviewers, including Jody C Culham as the Reviewing Editor and Reviewer #1, and the evaluation has been overseen by David Van Essen as the Senior Editor. The following individual involved in review of your submission has agreed to reveal their identity: Jonathan Smallwood (Reviewer #2).

The reviewers have discussed the reviews with one another and the Reviewing Editor has drafted this decision to help you prepare a revised submission.

Summary:

The reviewers agreed that your paper was an elegant and well-conducted study showing that disambiguation of Mooney images makes their neural representations more similar to the respective unambiguous images and less similar to one another. These results are observed not only in visual regions but also the frontoparietal and default mode networks and suggest richer perceptual representations than may have been expected. A particular strength of the experiment is the very clever method employed which enables the differentiation between meaningful and non meaningful stimuli in a manner that is not confounded with differences in stimulus properties. A fascinating aspect of the results is that they show regions in transmodal cortex, including the default mode network, play an important role in the processing of meaning from the external information. Moreover, while other studies have examined the transitions from analyzing perceptual features in visual areas to semantic features in higher areas, the absence of any particular semantic task other than naming suggests that fronto-parietal and default mode networks access semantics by default.

Essential revisions:

Two substantive comments were raised by the reviewers:

1) Two reviewers raised concerns about the inability to differentiate disambiguation from repetition suppression. This concern will need to be addressed by additional analysis or perhaps stronger argumentation.

One reviewer stated:

I have one substantive concern. The paradigm relies on showing participants a set of Mooney images before and after disambiguation and examining representational similarity. However, given the phenomenon of repetition suppression (or adaptation or priming), it is difficult to discern the degree to which representational changes are due to disambiguation vs. mere repetition. The ideal solution would have been to include control images that were presented the same number of times but never disambiguated. In the absence of that, perhaps other analyses could be done to show that effects between Pre and Post are not observed between early Pre and late Pre or between early Post and late Post.

Another reviewer stated:

A major concern is that at least the results reported in Figures 3 and 4 are explainable by neural adaptation rather than the transition from uninterpreted to interpreted. The most clear case is the greater dissimilarity of Post and Gray images compared to Pre images (Figures 3C and 4C). Adaptation between two images will reduce activity of the neural populations they have in common and leave the contrastive neural populations that do not commonly represent them. The Post Mooney images receive adaptation from the contrasting Post images in their run, the same contrasting images in the previous run (which were Pre images at the time), and presumably quite a bit of adaptation from the contrasting Gray images as well. The relevant Pre images from the previous run had only received adaptation from the contrasting Pre images that shared the same run. Thus the non-contrastive neurons representing the Post images should have been much more adapted than the non-contrastive neurons representing the Pre images, predicting greater dissimilarity among the Post images than the Pre images (Figures 3D and 4D). It is likely that the same could be said for the Gray images. To the degree that similar contrastive neurons are preserved in the Gray and Post images, it seems at least plausible that the same adaptation effects could make the Gray and Post images more similar to each other than they are the Pre images (Figures 3E and 4E). Regarding whether these same effects would increase dimensionality of the MDS (Figure 6), I don't know. It would certainly "sparsen" the patterns, which would seem to suggest the opposite reduction of dimensionality rather than the increase that was observed, but I think that is speculative when talking about such big complicated patterns.

2) One reviewer raised concerns about non-independence issues in the selection of FPN and DMN ROIs. The strongest rebuttal here would be a demonstration that the results hold with a fully independent definition of these ROIs (e.g., based on anatomy, Neurosynth, etc.) At minimum, the issue must be discussed and defended.

One possible concern is the rather piecemeal methods of selecting the FPN and DMN ROIs. The DMN was selected from a GLM contrast map (Post recognized vs. Pre recognized) and the FPN was selected from a whole brain SVM contrast between the same conditions. Because they didn't localize the networks with an independent resting state localizer, I'm assuming that these ROIs fell out of various explorations of the data. As far as I can tell, only one analysis in the paper is directly contaminated by double dipping. Figure 4D shows that similarity between Pre and Post Mooney images is less than similarity between Post and Gray scale. Similarity between Pre and Post (third bar) is artificially decreased because the FPN was selected using the SVM contrasting Pre and Post, selecting the regions where these conditions are least similar. The GLM contrast used to localize the DMN might also decrease the similarity between Pre and Post, but I'm not as sure about that one – maybe an RSA maven could help me out. It would at least be good to discuss these issues in the paper: 1) why were different contrasts required to locate the DMN and FPN and 2) acknowledge double dipping in Figure 4D. I think the paper would be strengthened by repeating the analyses using anatomical definitions of DMN and FPN.

[Editors' note: further revisions were requested prior to acceptance, as described below.]

Thank you for submitting your article "Content-specific activity in frontoparietal and default-mode networks during prior-guided visual perception" for consideration by *eLife*. Your resubmission has been reviewed by the Reviewing Editor, and the evaluation has been overseen by Eve Marder as the Senior Editor.

Although the reply to reviewers satisfactorily addressed the concerns raised by the reviewers, track changes showed no changes based on the major concerns. The two major concerns should be addressed directly in the manuscript. This is particularly true for the first comment – if two reviewers raised the same point, it could very well be a concern of other readers. The responses in the text do not have to be extensive as in the reply (but at least a brief statement addressing the crux of the concerns should appear in the main manuscript).

Once this correction has been made, the manuscript can proceed promptly to publication.

---

## [Author Response]

Essential revisions:Two substantive comments were raised by the reviewers:1) Two reviewers raised concerns about the inability to differentiate disambiguation from repetition suppression. This concern will need to be addressed by additional analysis or perhaps stronger argumentation.One reviewer stated:I have one substantive concern. The paradigm relies on showing participants a set of Mooney images before and after disambiguation and examining representational similarity. However, given the phenomenon of repetition suppression (or adaptation or priming), it is difficult to discern the degree to which representational changes are due to disambiguation vs. mere repetition. The ideal solution would have been to include control images that were presented the same number of times but never disambiguated. In the absence of that, perhaps other analyses could be done to show that effects between Pre and Post are not observed between early Pre and late Pre or between early Post and late Post.Another reviewer stated:A major concern is that at least the results reported in Figures 3 and 4 are explainable by neural adaptation rather than the transition from uninterpreted to interpreted. The most clear case is the greater dissimilarity of Post and Gray images compared to Pre images (Figures 3C and 4C). Adaptation between two images will reduce activity of the neural populations they have in common and leave the contrastive neural populations that do not commonly represent them. The Post Mooney images receive adaptation from the contrasting Post images in their run, the same contrasting images in the previous run (which were Pre images at the time), and presumably quite a bit of adaptation from the contrasting Gray images as well. The relevant Pre images from the previous run had only received adaptation from the contrasting Pre images that shared the same run. Thus the non-contrastive neurons representing the Post images should have been much more adapted than the non-contrastive neurons representing the Pre images, predicting greater dissimilarity among the Post images than the Pre images (Figures 3D and 4D). It is likely that the same could be said for the Gray images. To the degree that similar contrastive neurons are preserved in the Gray and Post images, it seems at least plausible that the same adaptation effects could make the Gray and Post images more similar to each other than they are the Pre images (Figures 3E and 4E). Regarding whether these same effects would increase dimensionality of the MDS (Figure 6), I don't know. It would certainly "sparsen" the patterns, which would seem to suggest the opposite reduction of dimensionality rather than the increase that was observed, but I think that is speculative when talking about such big complicated patterns.

We thank the reviewers for raising this important point. In order to control for the effect of repetition, we repeated the 1^st^-order RSA analysis while considering the subjects’ recognition status for each Mooney image in each presentation stage. First, for each subject, we selected the set of Mooney images that were unrecognized in the pre-disambiguation stage and yet recognized in the post-disambiguation stage (“Disambiguation set”), similar to the disambiguation effect assessed in the GLM and MVPA analyses shown in Figure 2. Second, for each subject, we selected the set of Mooney images that were *recognized* in both pre- and post-disambiguation stages (“Repetition set”). Importantly, in both cases, for each participant, the same set of Mooney images were used for Pre and Post conditions (group-mean and s.d. for the number of distinct Mooney images included in each set: Disambiguation set, 14 ± 3; Repetition set, 14 ± 5). Thus, the repetition set includes identical Mooney images that were recognized in both pre- and post-disambiguation stages, whereas the disambiguation set includes identical Mooney images that underwent a salient change in recognition status before vs. after disambiguation. *Since both sets of images underwent identical sequence of repetition, this allowed us to test whether the change in recognition status had an effect above and beyond that of repetition on the neural activity*.

To this end, we performed the following repeated-measures ANOVAs with three factors: Set (image-set type: Disambiguation vs. Repetition), Condition (Pre- vs. Post-disambiguation presentation stage), and Network (Visual, LOC, FG, FPN, and DMN, as in Figure 6D). The dependent variables were measures from 1^st^-order RSA as assessed in Figure 3D-E and Figure 4C-D. Before being entered in the ANOVA, r values of the RDMs were transformed to z-values, using the Fisher’s transform.

First, we compared the increase in between-image dissimilarity from Pre to Post stage (cyan brackets in Figures 3 and 4) in both data sets. This ANOVA revealed a clear effect of Condition (F_1,14_ = 24.3, *p* < 0.001), and, crucially, a Set x Condition interaction (F_1,14_ = 4.6, *p* = 0.05). Post-hoc comparisons revealed that the Pre-to-Post increase in dissimilarity was stronger in the Disambiguation Set (F_1_ = 24.9), compared to the Recognition set (F_1_ = 10.7).

Second, when comparing the increase in similarity from Pre-Gray to Post-Gray (red brackets in Figures 3 and 4), the same interaction was found (Set x Condition, F_1,14_ =14.9, *p* = 0.002). Again, post-hoc comparisons revealed the increase was larger in the Disambiguation set (F_1_ = 96.3) than in the Repetition set (F_1_ = 18.9).

Third, when comparing the difference in similarity between Pre-Post and Post-Gray (green brackets in Figures 3 and 4), we found the same interaction effect (Set x Condition, F_1,14_ = 6.3, *p* = 0.02). As in the previous analyses, the difference between Pre-Post and Post-Gray was larger in the Disambiguation set (F_1_ = 71.2) than in the Repetition set (F_1_ = 25.6).

Together, these analyses reveal that, as suggested by the reviewers, repetition has an effect on neural activity. *However, they demonstrate that disambiguation (i.e., change in recognition status of Mooney images due to viewing the corresponding Gray-scale images) influences neural representations above and beyond effects expected by mere repetition.*

Last, the first reviewer mentioned the interesting idea of including control images that were presented the same number of times but never disambiguated. Due to the long duration of our experiment, which was already pushing the limit of subjects’ tolerability in the 7T scanner (~2 hr), we could not include control images in this experiment. However, we would like to mention that in a separate magnetoencephalography (MEG) experiment from our lab (Flounders et al., in prep), using a similar paradigm and the same 33 Mooney images as studied here, plus control Mooney images that were presented in the same manner but shown with non-matching gray-scale images (and thus never disambiguated), we found a similar pattern of disambiguation-related effects on both behavior and neural activity in the regular images, which did not extend to control images.

2) One reviewer raised concerns about non-independence issues in the selection of FPN and DMN ROIs. The strongest rebuttal here would be a demonstration that the results hold with a fully independent definition of these ROIs (e.g., based on anatomy, Neurosynth, etc.) At minimum, the issue must be discussed and defended.One possible concern is the rather piecemeal methods of selecting the FPN and DMN ROIs. The DMN was selected from a GLM contrast map (Post recognized vs. Pre recognized) and the FPN was selected from a whole brain SVM contrast between the same conditions. Because they didn't localize the networks with an independent resting state localizer, I'm assuming that these ROIs fell out of various explorations of the data. As far as I can tell, only one analysis in the paper is directly contaminated by double dipping. Figure 4D shows that similarity between Pre and Post Mooney images is less than similarity between Post and Gray scale. Similarity between Pre and Post (third bar) is artificially decreased because the FPN was selected using the SVM contrasting pre and post, selecting the regions where these conditions are least similar. The GLM contrast used to localize the DMN might also decrease the similarity between Pre and Post, but I'm not as sure about that one – maybe an RSA maven could help me out. It would at least be good to discuss these issues in the paper: 1) why were different contrasts required to locate the DMN and FPN 2) acknowledge double dipping in Figure 4D. I think the paper would be strengthened by repeating the analyses using anatomical definitions of DMN and FPN.

We have now repeated the analyses shown in Figure 4 using FPN and DMN ROIs defined in an independent resting-state data set (Power et al., 2011; as shown in Figure2—figure supplement 4D). The new results are presented in Author response image 1 and are nearly identical to the original results in Figure 4. This figure includes right-hemisphere and mid-line ROIs; results from left-hemisphere ROIs were also nearly identical to the original results shown in Figure4—figure supplement 1. Although not requested by the reviewer, we have also double checked the results presented in Figures 5, 6, 7 using these independently defined FPN and DMN ROIs, all of which were very similar to the original results.

Given the above results, and since both FPN and DMN are functionally heterogeneous networks (e.g., Andrews-Hanna et al., 2010), we have opted to retain the original figures in the manuscript using ROIs defined from independent analyses on the same data set, which should extract regions that are more specifically involved in this task.

To answer the reviewer’s question about why different contrasts were used to extract DMN and FPN ROIs: as we mention in the text, changes in the overall activation magnitude between pre- and post-disambiguation Mooney images were previously reported in DMN regions (Dolan et al., 1997; Gorlin et al., 2012), and thus this finding was expected. By contrast, MVPA probes voxel-wise activation pattern – information that is complementary to the overall activation magnitude. Using MVPA, we found significant decoding of presentation stage (pre- vs. post-disambiguation) in regions of the FPN – a novel finding of the present study. Neither of these two analyses investigated information encoding specific to individual images, which was the subject of the following RSA analyses.

[Editors' note: further revisions were requested prior to acceptance, as described below.]

Although the reply to reviewers satisfactorily addressed the concerns raised by the reviewers, track changes showed no changes based on the major concerns. The two major concerns should be addressed directly in the manuscript. This is particularly true for the first comment – if two reviewers raised the same point, it could very well be a concern of other readers. The responses in the text do not have to be extensive as in the reply (but at least a brief statement addressing the crux of the concerns should appear in the main manuscript.Once this correction has been made, the manuscript can proceed promptly to publication.

We have now addressed both major concerns in the main text of the manuscript – major concern #1: subsection “Disambiguation shifts neural representations toward the prior throughout the cortical hierarchy”; major concern #2: subsections “Disambiguation leads to widespread changes in neural activity” and “Region-of-interest (ROI) definition”.